# What functions can Graph Neural Networks compute on random graphs? The role of Positional Encoding

**Nicolas Keriven**
CNRS, IRISA, Rennes, France
`nicolas.keriven@cnrs.fr`

**Samuel Vaiter**
CNRS, LJAD, Nice, France
`samuel.vaiter@cnrs.fr`

## Abstract

We aim to deepen the theoretical understanding of Graph Neural Networks (GNNs) on large graphs, with a focus on their expressive power. Existing analyses relate this notion to the graph isomorphism problem, which is mostly relevant for graphs of small sizes, or studied graph classification or regression tasks, while prediction tasks on *nodes* are far more relevant on large graphs. Recently, several works showed that, on very general random graphs models, GNNs converge to certains functions as the number of nodes grows. In this paper, we provide a more complete and intuitive description of the function space generated by equivariant GNNs for node-tasks, through general notions of convergence that encompass several previous examples. We emphasize the role of input node features, and study the impact of *node Positional Encodings* (PEs), a recent line of work that has been shown to yield state-of-the-art results in practice. Through the study of several examples of PEs on large random graphs, we extend previously known universality results to significantly more general models. Our theoretical results hint at some normalization tricks, which is shown numerically to have a positive impact on GNN generalization on synthetic and real data. Our proofs contain new concentration inequalities of independent interest.

## 1  Introduction

Machine learning on graphs with Graph Neural Networks (GNNs) [53, 5] is now a well-established domain, with application fields ranging from combinatorial optimization [6] to recommender systems [50, 11], physics [45, 1], chemistry [16], epidemiology [37], physical networks such as power grids [41], and many more. Despite this, there is still much that is not properly understood about GNNs, both empirically and theoretically, and their performances are not always consistent [52, 22], compared to simple baselines in some cases. It is generally admitted that a better theoretical understanding of GNNs, especially of their fundamental limitations, is necessary to design better models in the future.

Theoretical studies of GNNs have largely focused on their *expressive power*, kickstarted by a seminal study [54] that relates their ability to *distinguish non-isomorphic graphs* to the historical Weisfeiler–Lehman (WL) test [51]. Following this, many works have defined improved versions of GNNs to be "more powerful than WL" [34, 35, 26, 49, 38], often by augmenting GNNs with various features, or by implementing "higher-order" versions of the basic message-passing paradigm. Among the simplest and most effective idea to "augment" GNNs is the use of *Positional Encodings* (PE) as input to the GNN, inspired by the vocabulary of Transformers [48]. The idea is to equip nodes with carefully crafted input features that would help break some of the indeterminancy in the subsequent message-passing framework. In early works, unique and/or random node identifiers have been used [32, 47], but they technically break the permutation-invariance/equivariance – consistency with a reordering of the nodes in the graph – of the GNN. Most PEs in the current literature are based on eigenvectors of the adjacency matrix or Laplacian of the graph [12, 13] (with recent variants to handle the sign/basis indeterminancy

37th Conference on Neural Information Processing Systems (NeurIPS 2023).

[31]), random-walks [13], node metrics [56, 30], or subgraphs [4]. Some of these have been shown to have an expressive power beyond WL [30, 4, 31].

In some contexts however, WL-based analyses have limitations: they pertain to tasks on graphs (e.g. graph classification or regression) and have limited to no connections to tasks on nodes; and they are mostly relevant for small-scale graphs, as *medium or large graphs have a negligible chance of being exactly isomorphic to one another*, but exhibit different characteristics (e.g. community structures) that might be useful for learning. At the other end of the spectrum, the properties of GNNs on *large* graphs have been analysed in the context of latent positions Random Graphs (RGs) [24, 25, 43, 44, 29, 2, 36], a family of models slightly more general than graphons [33]. Such statistical models of large graphs are classically used in graph theory [18, 9] to model various data such as epidemiological [27, 39], biological [17], social [20], or protein-protein interaction [19] networks, and are still an active area of research [9]. For GNNs, the use of such models has shed light on their stability to deformations of the model [44, 29, 24], expressive power [25], generalisation [15, 36], or some phenomena such as oversmoothing [23, 3]. One basic idea is that, as the number of nodes in a random graph grows, GNNs converge to "continuous" equivalents [24, 8], whose properties are somewhat easier to characterize than their discrete counterpart. As prediction tasks on *nodes* are far more common and relevant on large graphs modelled by random graphs, this paper will focus on *permutation-equivariant* GNNs, rather than permutation-invariant. In the limit, it has been shown that their output converge to *functions* over some latent space to label the nodes, but the descriptions of this space of functions and its properties are still very much incomplete. A partial answer was given in [25], in which some universality properties are given for specific models of GNNs, but for limited models of random graphs *with no random edges*, and specific models of GNNs that no not include node features or PEs. We will in particular extend some of their results to random edges, with the proper choice of PEs.

**Contributions.** In this paper, we significantly extend existing results by providing a **complete description of the function space generated by permutation-equivariant GNNs** (Theorem 1), in terms of simple stability rules, and show that it is equivalent to previous implicit definitions that were based on convergence bounds. We outline the **role of the input node features**, and particularly of Positional Encodings (PEs). We then study the several representative examples of PEs on large random graphs. In particular, we analyze SignNet [31] (eigenvector-based) PEs (Theorem 2), and distance-based PEs [30] (Theorem 3). We derive simple normalization rules that are necessary for convergence, and illustrate them on real data. Finally, our proofs contain new universality results for square-integrable functions and new concentration inequalities that are of independent interest. All technical proofs are available in the Appendix. The code to reproduce the figures can be found at https://github.com/nkeriven/random-graph-gnn.

## 2 Background on Random Graphs and Graph Neural Networks

Let us start with generic notations and definitions. The norm $\|\cdot\|$ is the Euclidean norm for vectors and the operator norm for matrices and compact operators between Hilbert spaces. The latent space $\mathcal{X}$ is a compact metric set with a probability distribution $P$ over it. Square-integrable functions from $\mathcal{X}$ to $\mathbb{R}^q$ w.r.t. $P$ are denoted by $L_q^2$, and are equipped with the Hilbertian norm $\|f\|_{L^2}^2 \overset{\text{def.}}{=} \int_{\mathcal{X}} \|f(x)\|^2 \, dP(x)$. The (disjoint) union of multidimensional functions $L_{\sqcup}^2 \overset{\text{def.}}{=} \bigsqcup_{q \in \mathbb{N}^*} L_q^2$ is a metric space for a metric defined as $\|f - g\|_{L^2}$ if $f, g \in L_q^2$ for some $q$, and 1 otherwise. Continuous *Lipschitz* functions between metric spaces $\mathcal{X} \to \mathcal{Y}$ are denoted by $\mathcal{C}_{\text{Lip}}(\mathcal{X}, \mathcal{Y})$. For $X = \{x_1, \dots, x_n\}$ where $x_i \in \mathcal{X}$, we define the sampling of $f : \mathcal{X} \to \mathbb{R}^d$ as $\iota_X f = [f(x_i)]_{i=1}^n \in \mathbb{R}^{n \times d}$. Given $Z \in \mathbb{R}^{n \times d}$, the Frobenius norm is $\|Z\|_{\text{F}}$ and we define the normalized Frobenius norm as $\|Z\|_{\text{MSE}} = n^{-\frac{1}{2}} \|Z\|_{\text{F}}$. The notation comes from the fact that $\|\iota_X(f - f^\star)\|_{\text{MSE}}^2 = n^{-1} \sum_i \|f(x_i) - f^\star(x_i)\|^2$ which is akin to a Mean Square Error.

**Latent position Random Graphs.** In this paper, we consider *latent position random graphs* [20, 28, 33], a family of models that includes Stochastic Block Models (SBM), graphons, random geometric graphs, and many other examples. They are the primary models used for the study of GNNs in the literature [24, 25, 29, 43]. We generate a graph $G = (X, A, Z)$, where $X \in \mathbb{R}^{n \times d}$ are *unobserved* latent variables, $A \in \{0, 1\}^{n \times n}$ its symmetric adjacency matrix, and $Z \in \mathbb{R}^{n \times p}$ are (optional) observed node features. The latent variables and adjacency matrix are generated as such:

$$\forall i, \ x_i \overset{iid}{\sim} P, \qquad \forall i < j, \ a_{ij} \sim \text{Bernoulli}(\alpha_n w(x_i, x_j)) \quad \text{independently} \tag{1}$$

where $w : \mathcal{X} \times \mathcal{X} \to [0,1]$ is a continuous *connectivity kernel* and $\alpha_n$ is the *sparsity-level* of the graph, such that the expected degrees are in $\mathcal{O}\left(n^2\alpha_n\right)$. Non-dense graph can be obtain with $\alpha_n = o(1)$, here we will go down to the *relatively sparse* case $\alpha_n \gtrsim (\log n)/n$, a classical choice in the literature [28, 24]. Note that the continuity hypothesis of the kernel $w$ is not really restrictive: neither $\mathcal{X}$ nor the support of the distribution $P$ need be connected. For instance, SBMs can be obtained by taking $\mathcal{X}$ to be a finite set. We do not specify a model for the node features yet, see Sec. 4.

**Graph shift matrix and operator.** When the number of nodes $n$ grows on random graphs, it is known that certain discrete operators associated to the graph converge to their continuous version, as well as the GNNs that employ them [24, 8]. Here, some of our results will be valid under quite general assumptions, hence we use generic notations for our graph representations. When the results are only valid for particular examples, this will be specifically expressed.

We consider a **graph shift matrix** [46] $S = S(G) \in \mathbb{R}^{n \times n}$, which can be either directly the adjacency matrix of the graph or various notions of graph Laplacians. We define an associated **graph shift operator** $\mathbf{S} : L^2_\sqcup \to L^2_\sqcup$ such that the restriction $\mathbf{S}_{|L^2_q}$ is a compact linear operator of $L^2_q$ onto itself. Note that we reserve "matrix" and "operator" respectively for the discrete and continuous versions. The results in Sec. 3 will be valid under generic convergence assumptions from $S$ to $\mathbf{S}$, while the results of Sec. 4 will focus on the following two representative examples.

**Example 1** (Normalized adjacency matrix and kernel operator). *Here $S = \tilde{A} = (n\alpha_n)^{-1}A$ and $\mathbf{S}f = \mathbf{A}f = \int w(\cdot, x)dP(x)$. This choice requires to know, or estimate, the sparsity level $\alpha_n$. In this case, our results will hold whenever $\alpha_n \gtrsim (\log n)/n$ with an arbitrary multiplicative constant.*

Note that this choice requires to know (or estimate) the parameter $\alpha_n$, otherwise we will not have convergence between $S$ and $\mathbf{S}$, which can be limiting. This is not the case of the next example.

**Example 2** (Normalized Laplacian matrix[1] and operator). *Here $S = L = D_A^{-1/2}AD_A^{-1/2}$ where $D_A = \mathrm{diag}(A1_n)$ is the degree matrix of $G$, and $\mathbf{S}f = \mathbf{L}f = \int \frac{w(\cdot,x)}{\sqrt{d(\cdot)d(x)}}dP(x)$ where $d(\cdot) = \int w(\cdot, x)dP(x)$ is the degree function. Whenever we opt for this choice, we assume that $d_{\min} \overset{\text{def.}}{=} \inf_\mathcal{X} d(x) > 0$, and our results will hold whenever $\alpha_n \geqslant C(\log n)/n$ with a multiplicative constant $C$ that depends (in a non-trivial way) on $w$, see Thm. 9 in App. D.*

To sometimes unify notations, when we adopt these examples, we define $w_{\mathbf{S}}$ such that $w_{\mathbf{S}}(x, y) = w(x, y)$ in the adjacency case and $w_{\mathbf{S}}(x, y) = \frac{w(x,y)}{\sqrt{d(x)d(y)}}$ in the normalized Laplacian case. Therefore for these two examples the continuous operator has a single expression $\mathbf{S}f = \int w_{\mathbf{S}}(\cdot, x)dP(x)$.

**Graph Neural Network.** As mentioned in the introduction, we focus on *equivariant* GNNs that can compute functions over *nodes*, as this makes the most sense on large graphs that RGs seek to model. Recall that we observe a graph shift operator $S$ and node features $Z \in \mathbb{R}^{n \times p}$, and we return a vector per nodes $\Phi(S, Z) \in \mathbb{R}^{n \times d_L}$. We adopt a traditional GNN that uses the graph shift matrix $S$: given input features $Z^{(0)} \in \mathbb{R}^{n \times d_0}$,

$$Z^{(\ell)} = \rho\left(Z^{(\ell-1)}\theta_0^{(\ell-1)} + SZ^{(\ell-1)}\theta_1^{(\ell-1)} + 1_n(b^{(\ell)})^\top\right) \in \mathbb{R}^{n \times d_\ell},$$

$$\Phi_\theta(S, Z^{(0)}) = Z^{(L-1)}\theta^{(L-1)} + 1_n(b^{(L)})^\top \tag{2}$$

where $\rho$ is the ReLU function applied element-wise, and $\theta_i^{(\ell)} \in \mathbb{R}^{d_\ell \times d_{\ell+1}}$, $b^{(\ell)} \in \mathbb{R}^{d_\ell}$ are learnable parameters gathered in $\theta \in \Theta$. We denote by $\Theta$ the set of all possible parameters. For all classic choices of $S$, our definition of GNNs are a special case of message-passing NN (MPNN), which can be defined with a more general "aggregation" function. For the two examples above (adjacency and Laplacian), the aggregation function used is a sum, or a normalized sum.

We note that here we employ the ReLU function as a non-linearity, as some of our results will use its specific properties. Multi-Layer Perceptrons (MLP, densely connected networks) using the ReLU activation, and with potentially more than one hidden layer, will be denoted by $f_\gamma^{\text{MLP}}$, where $\gamma$ gathers their parameters.

---

[1]Note that the normalized Laplacian is traditionally defined as $\mathrm{Id} - L$, here it does not change our definition of GNNs since they include residual connections

Following recent literature [12, 13], we consider inputing *Positional Encoding* (PE) at each node. Such PE are generally computed using only the graph structure and concatenated to existing node features $Z$, here we simply introduce a generic notation:

$$Z^{(0)} = \text{PE}_\gamma(S, Z) \in \mathbb{R}^{n \times d_0} \tag{3}$$

with some parameter $\gamma \in \Gamma$. In our notations, the PE module uses the node features $Z$, generally by concatenating them to its output. For short, we may denote the whole architecture with PE and GNN as $\Phi_{\theta,\gamma}(S, Z) \overset{\text{def.}}{=} \Phi_\theta(S, \text{PE}_\gamma(S, Z))$. It is not difficult to see that if the PE computation is equivariant, then the whole GNN is equivariant: denoting by $\sigma$ a permutation matrix of $\{1, \ldots, n\}$,

$$\forall \sigma, \ \Phi_{\theta,\gamma}(\sigma S \sigma^\top, \sigma Z) = \sigma \Phi_{\theta,\gamma}(S, Z) \quad \Leftrightarrow \quad \forall \sigma, \ \text{PE}_\gamma(\sigma S \sigma^\top, \sigma Z) = \sigma \text{PE}_\gamma(S, Z).$$

All the examples of PEs examined in Sec. 4 are equivariant.

## 3   Function spaces of Graph Neural Networks

In this section, we provide a complete and intuitive description of the function space approximated by equivariant GNNs applied on RGs. All technical proofs are provided in App. A. It has been shown [24, 25, 8, 29] that GNNs converge to functions over the latent space: when the node features are a sampling of a certain function $\iota_X f^{(0)}$, then the output of the GNN is close to being a sampling of another function $\iota_X f^{(L)}$. Assuming the node features or PEs approximate some function set $\mathcal{B} \subset L_{\sqcup}^2$, we define the space of functions that a GNN can approximate as follows.

**Definition 1.** *Given a base set $\mathcal{B} \subset L_{\sqcup}^2$, the **set of functions approximated by GNNs** $\mathcal{F}_{\text{GNN}}(\mathcal{B})$ is formed by all the functions $f \in L_{\sqcup}^2$ such that: for all $\varepsilon > 0$, there are $\theta \in \Theta$, $f^{(0)} \in \mathcal{B}$ such that*

$$\mathbb{P}\Big( \big\| \Phi_\theta(S, \iota_X f^{(0)}) - \iota_X f \big\|_{\text{MSE}} \geqslant \varepsilon \Big) \xrightarrow[n \to \infty]{} 0. \tag{4}$$

In other words, $\mathcal{F}_{\text{GNN}}(\mathcal{B})$ are the functions whose sampling can be $\varepsilon$-approximated by the output of a GNN, with probability going to 1 as $n$ grows. Note that if the quantifiers of $\theta$, $f^{(0)}$ and $\varepsilon$ were reversed, the MSE would converge to 0 in probability. Here this is *not* the case: $\theta, f^{(0)}$ *may depend on* $\varepsilon$, which is akin to an approximation level. Similar to the permutation equivariance of GNNs, there is a notion of continuous equivariance for functions well-approximated by GNNs [24, 25, 8], where the permutations are replaced by bijections over the latent space $\mathcal{X}$. We adopt the notations $\mathcal{F}_{\text{GNN}}(\mathcal{B}) = \mathcal{F}_{\text{GNN}}(\mathcal{B}, w, P)$. For all continuous bijections $\phi$ over $\mathcal{X}$, we define $w_\phi(x, y) = w(\phi(x), \phi(y))$, $P_\phi = \phi^{-1} \sharp P$ where $\sharp$ is the push-forward operation, and $\mathcal{B}_\phi = \{f \circ \phi \mid f \in \mathcal{B}\}$. Then, we have the following result.

**Proposition 1.** *Let $S = S(A)$ be a graph shift operator that only depends on the adjacency matrix of the graph in a permutation-equivariant manner. Then, for all continuous bijections $\phi : \mathcal{X} \to \mathcal{X}$,*

$$\mathcal{F}_{\text{GNN}}(\mathcal{B}_\phi, w_\phi, P_\phi) = \{f \circ \phi \mid f \in \mathcal{F}_{\text{GNN}}(\mathcal{B}, w, P)\}.$$

That is, if one "permutes" the kernel $w$, the distribution $P$ and the base set $\mathcal{B}$, then the function space $\mathcal{F}_{\text{GNN}}$ contains exactly the permuted version of the original space.

The goal of this section is to provide a more intuitive description of the space $\mathcal{F}_{\text{GNN}}$, which we will do under some basic convergence assumption from $S$ to $\mathbf{S}$. GNNs (2) basically include two components: dense connections and MLPs that can approximate any continuous function by the universality theorem [40], and applications of $S$. Hence, we define the following function space.

**Definition 2.** *We define $\mathcal{F}_{\mathbf{S}}(\mathcal{B}) \subset L_{\sqcup}^2$ the (minimal) **S-extension** of a base set $\mathcal{B} \subset L_{\sqcup}^2$ by the following rules:*

(i) ***Base space:*** *$\mathcal{B} \subset \mathcal{F}_{\mathbf{S}}(\mathcal{B})$;*

(ii) ***Stability by composition with continuous functions:*** *for all $f \in \mathcal{F}_{\mathbf{S}}(\mathcal{B})$ with a $p$-dimensional output and $g \in \mathcal{C}_{\text{Lip}}(\mathbb{R}^p, \mathbb{R}^q)$, it holds[2] that $g \circ f \in \mathcal{F}_{\mathbf{S}}(\mathcal{B})$;*

(iii) ***Stability by graph operator:*** *for all $f \in \mathcal{F}_{\mathbf{S}}(\mathcal{B})$, it holds that $\mathbf{S}f \in \mathcal{F}_{\mathbf{S}}(\mathcal{B})$;*

---

[2]Note that, since $g$ is Lipschitz, when $f \in L_{\sqcup}^2$ we indeed have $g \circ f \in L_{\sqcup}^2$.

*(iv)* **Linear span:** *for all $q$, $\mathcal{F}_{\mathbf{S}}(\mathcal{B}) \cap L_q^2$ is a vector space;*

*(v)* **Closure:** *$\mathcal{F}_{\mathbf{S}}(\mathcal{B})$ is closed in $L_{\sqcup}^2$;*

*(vi)* **Minimality:** *for all $\mathcal{G} \subset L_{\sqcup}^2$ satisfying all the properties above, $\mathcal{F}_{\mathbf{S}} \subset \mathcal{G}$.*

In words, $\mathcal{F}_{\mathbf{S}}$ take a base set $\mathcal{B}$, and extend it to be stable by composition with Lipschitz functions, application of the graph operator, and linear combination (of its elements with the same dimensionality). Our result will use the following assumption, which is naturally true for our running examples.

**Assumption 1.** *With probability going to $1$, $\|S\|$ is bounded. Moreover, for all $f \in L_{\sqcup}^2$,*

$$\|S\iota_X f - \iota_X \mathbf{S} f\|_{\mathrm{MSE}} \xrightarrow[n \to \infty]{\mathcal{P}} 0$$

*where $\xrightarrow{\mathcal{P}}$ indicates convergence in probability.*

**Proposition 2.** *Assumption 1 is true for the adjacency matrix (ex. 1) and normalized Laplacian (ex. 2).*

Under this assumption, the main result of this section states that the functions well-approximated by GNNs are exactly the $\mathbf{S}$-extension of the base input features $\mathcal{B}$.

**Theorem 1.** *Under Assumption 1, for all $\mathcal{B} \subset L_{\sqcup}^2$, we have:*

$$\mathcal{F}_{\mathrm{GNN}}(\mathcal{B}) = \mathcal{F}_{\mathbf{S}}(\mathcal{B})$$

Given the definition of GNNs (2) and construction of $\mathcal{F}_{\mathbf{S}}$, Theorem 1 appears quite natural. Its proof, provided in App. A.3, is however far from trivial. The inclusion $\mathcal{F}_{\mathbf{S}}(\mathcal{B}) \subset \mathcal{F}_{\mathrm{GNN}}(\mathcal{B})$ is similar in spirit to previous convergence results [24], since one has to construct a GNN that approximates a particular function. It involves however a new extended universality theorem for MLPs for square-integrable functions (Lemma 3 in App. A.3), which uses *the special properties of ReLU*. The reverse inclusion $\mathcal{F}_{\mathrm{GNN}}(\mathcal{B}) \subset \mathcal{F}_{\mathbf{S}}(\mathcal{B})$ is quite different from previous work on GNN convergence: given $f \in \mathcal{F}_{\mathrm{GNN}}(\mathcal{B})$ whose only property is to be well-approximated by GNNs, one must construct a sequence of functions in $\mathcal{F}_{\mathbf{S}}(\mathcal{B})$ that converge to $f$, and uses the closure of $\mathcal{F}_{\mathbf{S}}(\mathcal{B})$. The need to work within square-integrable function is here obvious, as we only have convergence of the MSE, an approximation of the $L^2$-norm. For instance, this inclusion would not be true in the space of continuous functions.

Using composition with continuous functions, if $\mathcal{F}_{\mathbf{S}}(\mathcal{B})$ contains a continuous bijection $\phi : \mathcal{X} \to \mathrm{Im}(\phi)$, then $\mathcal{F}_{\mathbf{S}}(\mathcal{B})$ contains all continuous functions, and by density all square integrable functions. That is, the equivariant GNNs are then **universal** over $\mathcal{X}$: they can generate any function to label the nodes. Another criterion using the Stone-Weierstrass theorem (e.g. [21]), similar to the proofs in [25], is the following.

**Proposition 3.** *Assume that for all $x \neq x'$ in $\mathcal{X}$, there is a continuous function $f \in \mathcal{F}_{\mathbf{S}}(\mathcal{B}) \cap \mathcal{C}_{\mathrm{Lip}}(\mathcal{X}, \mathbb{R})$ such that $f(x) \neq f(x')$. Then, $\mathcal{F}_{\mathbf{S}}(\mathcal{B}) = L_{\sqcup}^2$.*

In the rest of the paper, we study several examples of PEs and corresponding set $\mathcal{B}$, that will generalize the results of [25]. We expect many other interesting characteristics of $\mathcal{F}_{\mathbf{S}}$ to be derived in the future.

## 4 Node features and Positional encodings

In the previous section, we have provided a complete description of the function space generated by equivariant GNNs when fed samplings of functions as node features, and the set of $\mathcal{B}$ is thus crucial for the properties of $\mathcal{F}_{\mathbf{S}}(\mathcal{B})$. For instance, in the absence of node features and PEs, it is classical to input *constant features* to GNNs [25], such that the space of interest is $\mathcal{F}_{\mathbf{S}}(1)$. However, similar to the failure of the WL test on regular graphs, if $\mathbf{S}1 \propto 1$ (e.g. constant degree function), then $\mathcal{F}_{\mathbf{S}}(1)$ *contains only constant functions*! The role of PEs is often to mitigate such situations.

**Definition 3.** *The **set of functions approximated by PEs** $\mathcal{F}_{\mathrm{PE}}$ is formed by all the functions $f \in L_{\sqcup}^2$ such that: for all $\varepsilon > 0$, there is $\gamma \in \Gamma$ such that*

$$\mathbb{P}\Big( \|\mathrm{PE}_\gamma(S, Z) - \iota_X f\|_{\mathrm{MSE}} \geqslant \varepsilon \Big) \xrightarrow[n \to \infty]{} 0. \tag{5}$$

Note that, as before, $\gamma$ may depend on $\varepsilon$. When passing PEs as input to GNNs, $\mathcal{F}_{\mathrm{PE}}$ serves as the base space $\mathcal{B}$, and the space of interest to characterize the functions well approximated by the whole

architecture $\Phi_{\theta,\gamma}$ is therefore $\mathcal{F}_{\mathbf{S}}(\mathcal{F}_{\mathrm{PE}})$. In fact, by simple Lipschitz property: for any $f \in \mathcal{F}_{\mathbf{S}}(\mathcal{F}_{\mathrm{PE}})$ and $\varepsilon > 0$, there are $\theta \in \Theta, \gamma \in \Gamma$ such that

$$\mathbb{P}\Big( \big\| \Phi_{\theta,\gamma}(S, Z) - \iota_X f \big\|_{\mathrm{MSE}} \geqslant \varepsilon \Big) \xrightarrow[n\to\infty]{} 0$$

In the rest of the section, we therefore aim to characterize $\mathcal{F}_{\mathrm{PE}}$ for several representative examples. We first briefly comment on observed node features, then move on to PEs. Proofs are in App. B.

### 4.1 Node features

A first, simple example, is when observed node features are actually a sampling of some function $Z = \iota_X f^{(0)}$. This is a convenient choice that is often adopted in the literature [24, 25, 23, 8, 29]. In this case, by adopting the identity $\mathrm{PE}_\gamma(S, Z) = Z$, it is immediate that $\mathcal{F}_{\mathrm{PE}} = \{f^{(0)}\}$. A more realistic example is the presence of centered noise:

$$Z = \iota_X f^{(0)} + \nu \in \mathbb{R}^{n \times d_0} \tag{6}$$

where $\nu = [\nu_1, \ldots, \nu_n]$ and the $\nu_i$ are i.i.d. noise vectors with $\mathbb{E}\nu_i = 0$ and $\mathrm{Cov}(\nu_i) = C_\nu$. This time, $\mathcal{F}_{\mathrm{PE}}$ cannot contain directly $f^{(0)}$, as the Law of Large Numbers (LLN) gives

$$\big\| Z - \iota_X f^{(0)} \big\|_{\mathrm{MSE}}^2 = \|\nu\|_{\mathrm{MSE}}^2 \xrightarrow[n\to\infty]{\mathcal{P}} \mathrm{Tr}(C_\nu) > 0$$

However, when applying the graph shift matrix at least once, one obtains convergent PEs.

**Proposition 4.** *Consider the adjacency matrix (ex. 1) or normalized Laplacian (ex. 2). If the node features are a noisy sampling (6) and the PE are defined $\mathrm{PE}_\gamma(S, Z) = SZ$, then, $\mathcal{F}_{\mathrm{PE}} = \{\mathbf{S}f^{(0)}\}$.*

Of course this may not be the only possibility for removing noise from node features, and moreover it is not clear how realistic the node features model (6) actually is. The study of more refined models linking graph structure and node features is a major path for future work.

### 4.2 Positional Encodings

In this section, we consider classical PEs computed solely from the graph structure and show how they articulate with our framework. We consider two examples that are the most-often used in the literature: PEs as eigenvectors of the graph shift matrix [12, 13] (actually a recent variant that account for sign indeterminancy [31]), and PE based on distance-encoding [30] (again a variant that, as we will see, generalize other architectures [49]). For most of the results below, we will focus on two representative cases of kernels, that include many practical examples. We remark that these yield *sufficient* conditions to establish our results, but by no mean necessary. Other cases could be examined in future work.

**Example a** (Stochastic Block Models). *In this case, the space of latent variables $\mathcal{X} = \{1, \ldots, K\}$ is finite, each element correspond to a community label. The kernel $w$ is represented by a matrix $C \stackrel{\text{def.}}{=} [w(\ell, k)] \in \mathbb{R}_+^{K \times K}$ that gives the probability of connection between communities $\ell$ and $k$, and $P \in \mathbb{R}_+^K$ is a probability vector of size $K$ that sum to $1$.*

**Example b** (P.s.d. kernel). *Here we assume that $w$ is positive semi-definite (p.s.d.). This includes for instance the Gaussian kernel.*

For any symmetric matrix (resp. self-adjoint compact operator) $M$, we denote by $\lambda_i^M$ its eigenvalues and $u_i^M$ its eigenvectors (resp. eigenfunctions), with any arbitrary choice of sign or basis here. Since in all our examples operators are either p.s.d. or finite-rank, the eigenvalues are ordered as such: first the non-zero eigenvalues by decreasing order (from positive to negative), then all zero eigenvalues.

#### 4.2.1 Eigenvectors and SignNet

It has been proposed [12, 13] to feed the first $q$ eigenvectors of the graph into the GNN, for a fixed $q$. A potential problem with this approach is the sign ambiguity of the eigenvectors, or even the basis ambiguity in case of eigenvalues with multiplicities. Here we consider only the sign ambiguity for simplicity: we will assume that the first eigenvalue of $\mathbf{S}$ are distinct. The sign ambiguity was alleviated in [31] by taking a *sign-invariant* function: considering an eigenvector $u_i^S$ of $S$,

$$(\mathbf{Q}f)(u_i^S) \stackrel{\text{def.}}{=} f(u_i^S) + f(-u_i^S) \in \mathbb{R}^{n \times p} \tag{7}$$

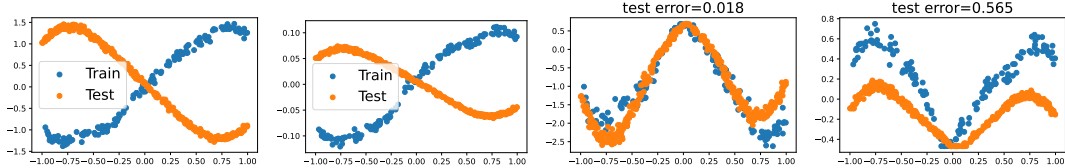

Figure 1: Illustration of the role of the SignNet architecture and of the renormalization by $\sqrt{n}$ of the eigenvectors on synthetic data, with a latent space $\mathcal{X} = [-1, 1]$ ($x$-axis), a Gaussian kernel $w$, and uniform distribution $P$. Blue dots represent a graph from the training set, orange dot a test graph that is twice bigger. **From left to right:** eigenvectors with renormalization (with a different sign for the two graphs), eigenvectors without, PEs with, and PEs without, with the regression test errors of a GNN trained using these PE with or without renormalization. We observe that SignNet indeed fixed the sign ambiguity. The absence of renormalization yields unconsistent PEs across graphs of different sizes, which results in a high test error on test graphs than training graphs.

where $f : \mathbb{R} \to \mathbb{R}^p$ is a function applied to each coordinate of $u_i^S$ to preserve permutation-equivariance. The resulting function is sign-invariant, and one can parameterized $f$. Given the first $q$ eigenvectors $u_i^S$ and a collection of MLPs $f_{\gamma_i}^{\text{MLP}} : \mathbb{R} \to \mathbb{R}^{p_i}$ for some output dimensions $p_i$, the PE considered in this subsection concatenates the outputs:

$$\text{PE}_\gamma(S) = [(\mathbf{Q} f_{\gamma_i}^{\text{MLP}})(\sqrt{n} u_i^S)]_{i=1}^q \in \mathbb{R}^{n \times p} \tag{8}$$

where $p = \sum_{i=1}^q p_i$ and the MLP are applied element-wise. The parameter $\gamma$ gathers the $\gamma_i$. The equation (8) involves a renormalization of the eigenvectors $u^S$ by the square root of the size of the graph $\sqrt{n}$: indeed, as $u_i^S$ is normalized *in* $\mathbb{R}^n$, this is necessary for consistency across different graph sizes. See Sec. 4.2.3 for a discussion and some numerical illustrations.

As can be expected, the eigenvectors of $S$ generally converge to the eigenfunctions of $\mathbf{S}$, under a spectral gap assumption. We provide the theorem below which handles all of our running examples. We suppose that the relevant eigenvalues have single multiplicities, to only have sign ambiguity.

**Theorem 2.** *Consider either SBM (ex. a) or p.s.d. kernel (ex. b), and either adjacency matrix (ex. 1) or normalized Laplacian (ex. 2). Fix $q$, assume the first $q+1$ eigenvalues $\lambda_1^{\mathbf{S}}, \ldots, \lambda_{q+1}^{\mathbf{S}}$ of $\mathbf{S}$ are two-by-two distinct. We define*

$$\mathcal{F}_{\text{Eig}} \stackrel{\text{def.}}{=} \left\{ [(\mathbf{Q} f_i) \circ u_i^{\mathbf{S}}]_{i=1}^q \mid f_i \in \mathcal{C}_{\text{Lip}}(\mathbb{R}, \mathbb{R}^{p_i}), p_i \in \mathbb{N}^* \right\} \tag{9}$$

*Then $\mathcal{F}_{\text{PE}} = \overline{\mathcal{F}_{\text{Eig}}}$.*

Hence $\mathcal{F}_{\text{PE}}$ contains the eigenfunctions of $\mathbf{S}$, modified by the SignNet architecture to account for the sign indeterminancy. We further discuss this space in Sec. 4.2.3. An illustration is provided in Fig. 1.

### 4.2.2  Distance-encoding PEs

In [30], the authors propose to define PEs through the aggregation of a set of "distances" $\xi(i, j)$ from each node $i$ to a set $j \in V_T$ of target nodes (typically, labelled nodes in semi-supervised learning, or anchor nodes selected randomly [56]):

$$(\text{PE}_\gamma)_{i,:} = \text{AGG}(\{\xi(i,j) \mid j \in V_T\})$$

where AGG is an *aggregation* function that acts on (multi-)sets, and $\xi(i, j)$ is selected in [30] as random-walk based distances $\xi(i, j) = [(AD_A^{-1})_{ij}, \ldots, ((AD_A^{-1})^q)_{ij}] \in \mathbb{R}^q$. For simplicity, since here we do not consider any particular set of target nodes, we just consider $V_T = V$ the set of all nodes. Moreover, to use our convergence results, we replace the random walk matrix with our graph shift matrix $S$. As aggregation, we opt for the deep-set architecture [58], which applies an MLP on each $\xi(i, j)$ then a sum. Deep sets can approximate any permutation-invariant function. As we will see below, with the proper normalization to ensure convergence, we obtain:

$$\text{PE}_\gamma = \frac{1}{n} \sum_j f_\gamma^{\text{MLP}} (n \cdot [Se_j, \ldots, S^q e_j]) \in \mathbb{R}^{n \times q}$$

where $f_\gamma^{\text{MLP}} : \mathbb{R}^q \to \mathbb{R}^p$ is applied row-wise and $e_j \in \mathbb{R}^n$ are one-hot basis vectors. We note that a similar architecture was proposed in a different line of work: it was called Structured Message Passing by [49], or Structured GNN by [25]. In these works, the inspiration is to give nodes unique identifiers,

*e.g.*, one-hot encodings $e_i$. However, this process is not equivariant. To restore equivariance, [49] propose a deep-set pooling in the "node-id" dimension $\mathrm{PE}_\gamma(S) = \sum_j \Phi_\gamma(S, e_j)$, where $\Phi_\gamma$ is itself a permutation-equivariant GNN, and the equivariance of $\mathrm{PE}_\gamma$ is restored. By choosing $\Phi_\gamma(\mathbf{S}, e_j) = n^{-1} f_\gamma^{\mathrm{MLP}}(n \cdot [Se_j, \dots, S^q e_j])$ (which is a valid choice for a message-passing GNN), we obtain exactly distance-encoding PEs above.

In [25], powerful universality results were shown for this choice of architecture *in the case of non-random edges* $a_{ij} = w(x_i, x_j)$ and $q = 1$. With our notations, they implicitly studied PE functions of the following form: $\int f(w(\cdot, x))dP(x)$. This allows to *modify the values of the kernel* before computing the degree function, and can therefore break potential indeterminancy such as constant degrees. Unfortunately, their proof technique and the concentration inequalities they use are *not true anymore for Bernoulli random edges*, which are far more realistic than deterministic weighted edges. Here we show that for a large class of kernels, concentration can be restored when we add an MLP filter on the eigenvalues of $S$ with ReLU. Our definition of distance-encoding PEs is therefore:

$$\mathrm{PE}_\gamma = \tfrac{1}{n} \sum_j f_{\gamma_1}^{\mathrm{MLP}}\left(n \cdot [S_{\gamma_2} e_j, \dots, S_{\gamma_2}^q e_j]\right) \tag{10}$$

where $S_{\gamma_2} \stackrel{\text{def.}}{=} h_{f_{\gamma_2}^{\mathrm{MLP}}}(S)$ is a filter that applies an MLP $f_{\gamma_2}^{\mathrm{MLP}}$ on the eigenvalues of $S$.

**Theorem 3.** *Consider either SBM (ex. a) or p.s.d. kernel (ex. b), and either adjacency matrix (ex. 1) or normalized Laplacian (ex. 2). Consider the PE (10). We define*

$$\mathcal{F}_{\mathrm{Dist}} \stackrel{\text{def.}}{=} \left\{ \int f([\mathbf{S}\delta_x(\cdot), \dots, \mathbf{S}^q \delta_x(\cdot)])dP(x) \mid f \in \mathcal{C}_{\mathrm{Lip}}([0,1]^q, \mathbb{R}^p), p \in \mathbb{N}^* \right\} \tag{11}$$

*where $\mathbf{S}\delta_x \stackrel{\text{def.}}{=} \{z \mapsto w_{\mathbf{S}}(z, x)\}$ by abuse of notation. Then $\mathcal{F}_{\mathrm{Dist}} \subset \mathcal{F}_{\mathrm{PE}}$.*

Note that here we only have an inclusion $\mathcal{F}_{\mathrm{Dist}} \subset \mathcal{F}_{\mathrm{PE}}$ instead of an equality as in Thm. 2: indeed, we show that the PE (10) can approximate functions in $\mathcal{F}_{\mathrm{Dist}}$, but they may converge to other functions. Nevertheless, as a consequence of our analysis, all the universality results of [25, Sec. 5.3] are valid with the choice of PE (10), see Appendix C for a reminder using our notations. This is a strict, and non-trivial improvement over [25], as their results were only derived for non-random edges. For this, Theorem 3 relies mostly on a new concentration inequality for Bernoulli matrices with ReLU filters in Frobenius norm, that we give below since it is of independent interest.

**Theorem 4.** *Consider either SBM (ex. a) or p.s.d. kernel (ex. b), and either adjacency matrix (ex. 1) or normalized Laplacian (ex. 2). Define the Gram matrix $W = [w_{\mathbf{S}}(x_i, x_j)/n]_{ij}$. For all $\varepsilon > 0$, there is an MLP filter $S_\gamma = h_{f_\gamma^{\mathrm{MLP}}}(S)$ such that*

$$\mathbb{P}(\|S_\gamma - W\|_{\mathrm{F}} \geqslant \varepsilon) \to 0.$$

The proof of this theorem, given in appendix B.3, is inspired by the so-called USVT estimator [7]. One notes that the use of an MLP graph filter is quite unconventional. A more classical choice is polynomial filters: this avoids the diagonalization of $S$ by computing $\sum_k a_k S^k$, it is for instance the basis for the ChebNet architecture [10]. For the purpose of Theorems 3 and 4, *polynomial filters do not work, and ReLU is of crucial importance*: indeed, we need the filter to zero-out $\mathcal{O}(n)$ eigenvalues *uniformly* in some interval $[-\tau, \tau]$. This cannot be done with polynomials with a fixed number of parameters and growing $n \to \infty$. On the other hand, when choosing $f$ as an MLP with ReLU, due to the shape of this non-linearity, $f_{\gamma_2}^{\mathrm{MLP}}$ can be *uniformly* 0 *on a whole domain*. Of course, polynomial filters offer great computational advantages, and perform well in practice, despite their flaw in our asymptotic analysis. Moreover, ReLU is technically non-differentiable. Designing filters that offer both computational advantages and exact approximation is still an open question. In practice, we observe that the ReLU-filter *does* learn to approximate its expected shape, when we minimize the reconstruction error $\|S_\gamma - W\|_{\mathrm{F}}$ on synthetic data where $W$ is known, see Fig. 2.

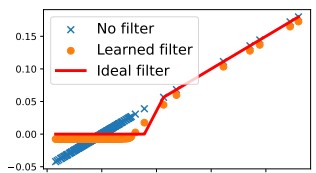

Figure 2: Illustration of Theorem 4 on synthetic data where $W$ is known, with a Gaussian kernel. Unfiltered eigenvalues of $S$ are represented by blue crosses, filtered ones obtained by minimizing $\min_{\gamma_2} \|S_{\gamma_2} - W\|_{\mathrm{F}}$ by orange dots, and the ideal ReLU-filter used in the proof of Thms. 3 and 4 is represented by a red line.

### 4.2.3 Discussion

**Approximation power.** As mentioned above, in the absence of node features, one may opt for constant input, but this may lead to degenerate situations. PEs aim to counteract that, by increasing GNNs' approximation power. We quickly verify that this is indeed the case for our two examples.

**Proposition 5.** *There are cases where $\mathcal{F}_{\mathbf{S}}(1) \subset \mathcal{F}_{\mathbf{S}}(\mathcal{F}_{\mathrm{Eig}})$ or $\mathcal{F}_{\mathbf{S}}(1) \subset \mathcal{F}_{\mathbf{S}}(\mathcal{F}_{\mathrm{Dist}})$ with strict inclusions.*

Moreover, as mentioned in the previous section, existing universality results [25] can be generalized in our case, see App. C. Another interesting question is somewhat the opposite: given the already rich class of functions generated by PEs, are GNNs really more powerful?

**Proposition 6.** *There are cases where $\mathcal{F}_{\mathrm{Eig}} \subset \mathcal{F}_{\mathbf{S}}(\mathcal{F}_{\mathrm{Eig}})$ or $\mathcal{F}_{\mathrm{Dist}} \subset \mathcal{F}_{\mathbf{S}}(\mathcal{F}_{\mathrm{Dist}})$, with strict inclusions.*

The proof, which is not so trivial, invokes functions with at least one round of message-passing after the computation of PEs, so the additional approximation power does not come only from MLPs. Intuitively, it seems natural that message-passing rounds are useful for other reasons, e.g. noise reduction or smoothing [23]. We leave these complementary lines of investigation for future work.

**Renormalization.** A striking point in our variants of PEs is the presence of various normalization factors by the graph size $n$ to ensure convergence: the equation (8) involves a renormalization of the eigenvectors $u^S$ by the square root of the size of the graph $\sqrt{n}$, while (10) involves a multiplicative factor $n$ *inside* the MLP $f_{\gamma_1}^{\mathrm{MLP}}$ (the $1/n$ outside of the sum is more classical). Our analysis shows that these normalization factors are necessary for convergence when $n \to \infty$, and more generally for consistency across different graph sizes.

In practice, this is generally not used. Indeed, if the training and testing graphs have roughly the same "range" of sizes $n \in [n_{\min}, n_{\max}]$, then a GNN model can *learn* the proper normalization to perform, which is not the point of view of our analysis $n \to \infty$. While in-depth benchmarking of PEs has been done in the literature [13] and is out-of-scope of this paper, we give a small numerical illustration of the effect of normalization in Table 1. We consider a synthetic dataset with a classic classification problem on (unobserved) latent variables $x_i$ and a Gaussian kernel $W$. To emphasize the effect of the normalization, we also examine a situation where the test graphs are much larger than the training graphs while following the same model, which we denote by `out-of-dist`. Concerning real data, since there are practically no datasets for node-classification with *several* graphs of sufficiently different size to test the renormalization, we artificially extract many subgraphs from a single large graph (`Citeseer`) with labelled nodes to create such a dataset, denoted by `Citeseer-subgraphs`. We also directly look at two graph-classification datasets with many graphs of different sizes. Note that, to emphasize the effect of PEs, we discard eventual node features and use only the graph structure.

On synthetic data exactly formed of random graphs of vastly different sizes, the renormalization is of course necessary to obtain good performance, as predicted by our theory: without it, the PEs do not converge when $n$ grows. On real data, we see that renormalization generally improve performance, and this is more true for IMDB-BINARY, which contains a larger range of graph sizes, and distance-based PEs. Note that here we use relatively small GNNs that are *not state-of-the-art* (in particular since we do not use node features), as well as a different train/test split than most papers ($K = 5$ CV-folds instead of $K = 10$): indeed, we do not want our models to *learn* the proper normalization on the limited range of sizes $n$ in the dataset, so we limit their number of parameters and use a smaller training set. We do not expect our simple renormalization process to make a significant difference on large-scale benchmarks with state-of-the-art models [13], but this is a pointer in an interesting direction to be explored in the future. In particular, this type of normalization may be useful in real-world scenarii where the test graphs are far larger than the labelled training graphs.

## 5 Conclusion

On large random graphs, the manner in which GNNs label *nodes* can be modelled by functions. The analysis of the resulting function spaces is still in its infancy, and of a very different nature to the studies of *graph-tasks*, both discrete [54] or in the limit [36]. In this paper, we clarified significantly the nature of the space of functions well-approximated by GNNs on large-graphs, showing that it can be defined by a few extension rules within the space of square-integrable functions. We then showed the usefulness of Positional Encodings by analyzing two popular examples, established new universality results, as

| Dataset | Eigenvectors | | Distance-encoding | |
| --- | --- | --- | --- | --- |
| | w/ norm. | w/o norm. | w/ norm. | w/o norm. |
| Synthetic | 68.61 | 65.59 | 67.31 | 62.49 |
| Synthetic (out-of-dist) | 67.87 | 62.51 | 66.80 | 63.33 |
| CiteSeer-subgraphs | 49.45 | 49.43 | 48.99 | 37.09 |
| IMDB-BINARY [55] (graph-classif.) | 67.80 | 66.10 | 71.10 | 63.95 |
| COLLAB [55] (graph-classif.) | 73.74 | 74.77 | 75.65 | 75.02 |

Table 1: Test accuracy for GNNs with different PEs, with or without renormalization by the graph size $n$. Results for 5-fold cross-validation averaged over 3 experiments.

well as some concentration inequalities of independent interest. Our theory hinted at some process for consistency across graphs of different sizes that can help generalization in practice.

This paper, which in large part consisted in *properly defining* the objects of interest, is without doubt only a first step in their analysis. Future studies might look at specific settings and derive more useful properties of the space $\mathcal{F}_{\mathbf{S}}$, more powerful PEs, a better understanding of their limitations, or more realistic models for node features. In particular, a better connection with the existing WL-based theory for *finite small* graphs, and associated "powerful" architectures, is a major path for future work.

## Acknowledgments

The authors acknowledge the fundings of ANR GraVa ANR-18-CE40-0005 and ANR GrandMa ANR-21-CE23-0006.

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

# A Proofs of Sec. 3

## A.1 Proof of Prop. 1

Denote $\mathcal{G}$ the distribution of $(X, A)$ with $(w, P)$ and $\mathcal{G}_\phi$ for $(w_\phi, P_\phi)$. Note that if $X, A \sim \mathcal{G}_\phi$, then $\phi(X), A \sim \mathcal{G}$, and recall that $S = S(A)$ only depends on $A$.

Consider $f \in \mathcal{F}_{\mathrm{GNN}}(\mathcal{B}, w, P)$, $\varepsilon > 0$, and $\theta, f^{(0)}$ such that $\mathbb{P}_\mathcal{G}\left( \left\| \Phi_\theta(S, \iota_X f^{(0)}) - \iota_X f \right\|_{\mathrm{MSE}} \geqslant \varepsilon \right) \to 0$. Then, denoting by $\phi(X) = \{\phi(x_1), \ldots, \phi(x_n)\}$,

$$\mathbb{P}_{\mathcal{G}_\phi}\left( \left\| \Phi_\theta(S, \iota_X(f^{(0)} \circ \phi)) - \iota_X(f \circ \phi) \right\|_{\mathrm{MSE}} \geqslant \varepsilon \right) = \mathbb{P}_{\mathcal{G}_\phi}\left( \left\| \Phi_\theta(S, \iota_{\phi(X)} f^{(0)}) - \iota_{\phi(X)} f \right\|_{\mathrm{MSE}} \geqslant \varepsilon \right)$$

$$= \mathbb{P}_\mathcal{G}\left( \left\| \Phi_\theta(S, \iota_X f^{(0)}) - \iota_X f \right\|_{\mathrm{MSE}} \geqslant \varepsilon \right) \to 0$$

which shows that $f \circ \phi \in \mathcal{F}_{\mathrm{GNN}}(\mathcal{B}_\phi, w_\phi, P_\phi)$ and one inclusion. The other inclusion is immediate by doing the same reasoning for $\phi^{-1}$.

## A.2 Proof of Prop. 2

Define $W = [w_\mathbf{S}(x_i, x_j)/n]$ the Gram matrix. Using Theorem 9, for both our examples we have

$$\|S - W\| \xrightarrow[n \to \infty]{\mathcal{P}} 0$$

Since $\|W\| \leqslant \sup_{x,y} |w_\mathbf{S}(x, y)|$ is bounded, it shows that $\|S\|$ is bounded with probability going to 1.

Let $f \in L_q^2$ and any $\varepsilon > 0$. Since continuous functions are dense in square-integrable functions on compact spaces (see e.g. [14, Sec. 8.2]), let $g \in \mathcal{C}_{\mathrm{Lip}}(\mathcal{X}, \mathbb{R}^q)$ such that $\|f - g\|_{L^2} \leqslant \varepsilon$. We have

$$\|W\iota_X g - \iota_X \mathbf{S}g\|_{\mathrm{MSE}}^2 = \frac{1}{n} \sum_i \left\| \frac{1}{n} \sum_j w_\mathbf{S}(x_i, x_j) g(x_j) - \int w_\mathbf{S}(x_i, x) g(x) dP(x) \right\|^2$$

$$\leqslant \left\| \frac{1}{n} \sum_j w_\mathbf{S}(\cdot, x_j) f(x_j) - \int w_\mathbf{S}(\cdot, x) g(x) dP(x) \right\|_\infty^2 \xrightarrow[n \to \infty]{\mathcal{P}} 0$$

where we have used Lemma 8 and the fact that $g$ is bounded.

Finally,

$$\|S\iota_X f - \iota_X \mathbf{S}f\|_{\mathrm{MSE}} \leqslant \|S\iota_X f - W\iota_X f\|_{\mathrm{MSE}} + \|W\iota_X f - W\iota_X g\|_{\mathrm{MSE}}$$
$$+ \|W\iota_X g - \iota_X \mathbf{S}g\|_{\mathrm{MSE}} + \|\iota_X \mathbf{S}g - \iota_X \mathbf{S}f\|_{\mathrm{MSE}}$$

Using $\|AB\|_\mathrm{F} \leqslant \|A\| \|B\|_\mathrm{F}$ and the LLN, and the fact that $\|f\|_{L^2}$, $\|W\|$, $\|\mathbf{S}\|$ are bounded, with probability going to 1,

$$\|S\iota_X f - \iota_X \mathbf{S}f\|_{\mathrm{MSE}} \leqslant \|S - W\| \|\iota_X f\|_{\mathrm{MSE}} + \|W\| \|\iota_X(f - g)\|_{\mathrm{MSE}}$$
$$+ \|W\iota_X g - \iota_X \mathbf{S}g\|_{\mathrm{MSE}} + \|\iota_X \mathbf{S}(g - f)\|_{\mathrm{MSE}}$$
$$\lesssim \|S - W\| \|f\|_{L^2} + \|W\| \|f - g\|_{L^2} + 0 + \|\mathbf{S}\| \|g - f\|_{L^2} \lesssim \varepsilon$$

which, since $\varepsilon$ was chosen arbitrarily, concludes the proof.

## A.3 Proof of Theorem 1

The proof uses intermediate results. Recall the definition of GNNs: given input node features $Z^{(0)} \in \mathbb{R}^{n \times d_0}$,

$$Z^{(\ell)} = \rho\left( Z^{(\ell-1)} \theta_0^{(\ell-1)} + SZ^{(\ell-1)} \theta_1^{(\ell-1)} + 1_n (b^{(\ell)})^\top \right) \in \mathbb{R}^{n \times d_\ell},$$

$$\Phi_\theta(S, Z^{(0)}) = Z^{(L-1)} \theta^{(L-1)} + 1_n (b^{(L)})^\top$$

We can define a continuous equivalent of GNNs, called c-GNNs in the literature [24], using the operator $\mathbf{S}$. Given $f^{(0)} \in L^2_{d_0}$,

$$f^{(\ell)} = \rho \left( (\theta_0^{(\ell-1)})^\top f^{(\ell-1)} + (\theta_1^{(\ell-1)})^\top \mathbf{S} f^{(\ell-1)} + b^{(\ell)} \right) \in L^2_{d_\ell},$$

$$\Phi_\theta(\mathbf{S}, f^{(0)}) = (\theta^{(L-1)})^\top f^{(L-1)} + b^{(L)}$$

Then, under our assumption on the operators $(S, \mathbf{S})$, discrete GNNs converge to continuous GNNs.

**Lemma 1.** *Suppose Assumption 1 holds. For all $f \in L^2_\sqcup$ and $\theta$,*

$$\|\Phi_\theta(S, \iota_X f) - \iota_X \Phi_\theta(\mathbf{S}, f)\|_{\mathrm{MSE}} \xrightarrow[n\to\infty]{\mathcal{P}} 0$$

*Proof.* Writing $Z^{(0)} = \iota_X f$ and $f^{(0)} = f$, we show by recursion on the layers that $\|Z^{(\ell)} - \iota_X f^{(\ell)}\|_{\mathrm{MSE}} \xrightarrow[n\to\infty]{\mathcal{P}} 0.$

For $\ell = 0$, we have exactly $\|Z^{(0)} - \iota_X f^{(0)}\|_{\mathrm{MSE}} = 0$. Assuming the convergence holds for $\ell - 1$, we have,

$$\begin{aligned}
\left\|Z^{(\ell)} - \iota_X f^{(\ell)}\right\|_{\mathrm{MSE}} &= \left\|\rho \left( Z^{(\ell-1)}\theta_0^{(\ell-1)} + S Z^{(\ell-1)}\theta_1^{(\ell-1)} + 1_n (b^{(\ell)})^\top \right)\right.\\
&\qquad\left. - \iota_X \rho \left( (\theta_0^{(\ell-1)})^\top f^{(\ell-1)} + (\theta_1^{(\ell-1)})^\top \mathbf{S} f^{(\ell-1)} + b^{(\ell)} \right) \right\|_{\mathrm{MSE}}\\
&\lesssim \left\|Z^{(\ell-1)}\theta_0^{(\ell-1)} + S Z^{(\ell-1)}\theta_1^{(\ell-1)} \right.\\
&\qquad\left. - (\iota_X f^{(\ell-1)})\theta_0^{(\ell-1)} + (\iota_X \mathbf{S} f^{(\ell-1)})\theta_1^{(\ell-1)}\right\|_{\mathrm{MSE}}\\
&\leqslant \left(\left\|\theta_0^{(\ell-1)}\right\| + \left\|\theta_1^{(\ell-1)}\right\|\|S\|\right) \left\|Z^{(\ell-1)} - \iota_X f^{(\ell-1)}\right\|_{\mathrm{MSE}}\\
&\qquad + \left\|(S\iota_X - \iota_X \mathbf{S})f^{(\ell-1)}\right\|_{\mathrm{MSE}}
\end{aligned}$$

using the Lipschitz property of $\rho$ in the first line, and $\|AB\|_{\mathrm{F}} \leqslant \|A\| \|B\|_{\mathrm{F}}$ after. The first term converges to 0 by recursion hypothesis since $\|S\|$ is bounded with probability going to 1, and the second converges to 0 by Assumption 1. This concludes the proof. □

**Lemma 2.** *Given a base space $\mathcal{B} \subset L^2_\sqcup$, denote by $\mathcal{F}_c(\mathcal{B}) \subset L^2_\sqcup$ the following space of all functions $f$ of the form:*

$$\begin{aligned}
&f^{(0)} \in \mathcal{B}\\
&f^{(\ell+1)} = g_1^{(\ell)} \circ f^{(\ell)} + g_2^{(\ell)} \circ \mathbf{S} f^{(\ell)} \qquad \text{where } g_1^{(\ell)}, g_2^{(\ell)} \in \mathcal{C}_{\mathrm{Lip}}(\mathbb{R}^{d_\ell}, \mathbb{R}^{d_{\ell+1}})\\
&f = f^{(L)} \in L^2_{d_L}
\end{aligned} \tag{12}$$

*for all $k, L, d_i$. Then $\mathcal{F}_c(\mathcal{B})$ is dense in $\mathcal{F}_{\mathbf{S}}(\mathcal{B})$.*

*Proof.* By definition, $\mathcal{F}_c \subset \mathcal{F}_{\mathbf{S}}$ since its contruction uses only rules that leave $\mathcal{F}_{\mathbf{S}}$ stable. Conversely, $\overline{\mathcal{F}_c}$ satisfies all the rules of stability of $\mathcal{F}_{\mathbf{S}}$ so by minimality $\mathcal{F}_{\mathbf{S}} \subset \overline{\mathcal{F}_c}$. □

**Lemma 3** (Universality in $L^2$). *Let $f \in L^2_q$ and $g \in \mathcal{C}_{\mathrm{Lip}}(\mathbb{R}^q, \mathbb{R}^p)$, for all $\varepsilon > 0$, there exists an MLP $f_\gamma^{\mathrm{MLP}}$ that uses ReLU, with two hidden layers, such that*

$$\left\|g \circ f - f_\gamma^{\mathrm{MLP}} \circ f\right\|_{L^2} \leqslant \varepsilon \tag{13}$$

*Proof.* Denote by $L_g$ the Lipschitz constant of $g$. Let $C_k = [-k, k]^q$, and $\mathcal{X}_k = \{x \in \mathcal{X} \mid f(x) \in C_k\}$, and $\xi_k = \int_{\mathcal{X}_k} \|f(x)\|^2 \, dP(x)$. We have $\xi_k$ positive and increasing, and $\lim_{k\to\infty} \xi_k = \|f\|^2_{L_2}$. Define $k_\varepsilon$ such that $\xi_{k_\varepsilon} \geqslant \|f\|^2_{L^2} - \frac{\varepsilon^2}{1+L_g^2+\|g(0)\|^2}$, such that $\int_{\mathcal{X}_{k_\varepsilon}^c} \|f(x)\|^2 \, dP(x) \leqslant \frac{\varepsilon^2}{1+L_g^2+\|g(0)\|^2}$.

Since $C_{k_\varepsilon}$ is compact, by the universality theorem [21, 40], there is an MLP $f_{\gamma'}^{\mathrm{MLP}}$ such that $\sup_{y \in C_{k_\varepsilon}} \left| g(y) - f_{\gamma'}^{\mathrm{MLP}}(y) \right| \leqslant \varepsilon$. Moreover, using the property of ReLU, it is easy to see that the following function can be implemented by an MLP:

$$f_{\gamma''}^{\mathrm{MLP}}(t) = \begin{cases} -k_\varepsilon & \text{for } t \leqslant -k_\varepsilon \\ t & \text{for } -k_\varepsilon \leqslant t \leqslant k_\varepsilon \\ k_\varepsilon & \text{for } t \geqslant k_\varepsilon \end{cases}$$

Then, we define $f_\gamma^{\mathrm{MLP}} = f_{\gamma''}^{\mathrm{MLP}} \circ f_{\gamma'}^{\mathrm{MLP}}$, where $f_{\gamma''}^{\mathrm{MLP}}$ is applied coordinate-wise. As a result, we have $f_\gamma^{\mathrm{MLP}}(y) = f_{\gamma'}^{\mathrm{MLP}}(y)$ on $C_{k_\varepsilon}$, and $\left\| f_\gamma^{\mathrm{MLP}}(y) \right\|_\infty \leqslant k_\varepsilon$ outside. Then, we have

$$\left\| g \circ f - f_\gamma^{\mathrm{MLP}} \circ f \right\|_{L^2}^2 = \int \left\| g \circ f(x) - f_\gamma^{\mathrm{MLP}} \circ f(x) \right\|^2 dP(x)$$

$$\leqslant \int_{\mathcal{X}_{k_\varepsilon}} \left\| g \circ f(x) - f_\gamma^{\mathrm{MLP}} \circ f(x) \right\|^2 dP(x)$$

$$+ 2 \left( \int_{\mathcal{X}_{k_\varepsilon}^c} \left\| g \circ f \right\|^2 dP(x) + \int_{\mathcal{X}_{k_\varepsilon}^c} \left\| f_\gamma^{\mathrm{MLP}} \circ f \right\|^2 dP(x) \right)$$

For the first term, since on $\mathcal{X}_{k_\varepsilon}$ we have $f(x) \in C_{k_\varepsilon}$, we use the approximation property and we have

$$\int_{\mathcal{X}_{k_\varepsilon}} \left\| g \circ f(x) - f_\gamma^{\mathrm{MLP}} \circ f(x) \right\|^2 dP(x) \leqslant \varepsilon^2$$

For the second term, since $\|f(x)\|^2 \geqslant dk_\varepsilon^2 \geqslant 1$ on $\mathcal{X}_{k_\varepsilon}^c$, we have

$$\int_{\mathcal{X}_{k_\varepsilon}^c} \|g \circ f\|^2 dP(x) \leqslant 2 \int_{\mathcal{X}_{k_\varepsilon}^c} \|g \circ f - g(0)\|^2 dP(x) + \|g(0)\|^2 \int_{\mathcal{X}_{k_\varepsilon}^c} 1 dP(x)$$

$$\leqslant 2(L_g^2 + \|g(0)\|^2) \int_{\mathcal{X}_{k_\varepsilon}^c} \|f\|^2 dP(x) \leqslant 2\varepsilon^2$$

And for the third term, given the property of the built MLP,

$$\int_{\mathcal{X}_{k_\varepsilon}^c} \left\| f_\gamma^{\mathrm{MLP}} \circ f \right\|^2 dP(x) \leqslant \int_{\mathcal{X}_{k_\varepsilon}^c} dk_\varepsilon^2 dP(x)$$

$$\leqslant \int_{\mathcal{X}_{k_\varepsilon}^c} \|f\|^2 dP(x) \leqslant \varepsilon^2$$

which concludes the proof. $\qquad\square$

**Lemma 4.** *Let $f \in \mathcal{F}_c(\mathcal{B})$. For all $\varepsilon > 0$, there exists $\theta$ and $f^{(0)} \in \mathcal{B}$ such that*

$$\left\| \Phi_\theta(\mathbf{S}, f^{(0)}) - f \right\| \leqslant \varepsilon \tag{14}$$

*Proof.* Let $f \in \mathcal{F}_c(\mathcal{B})$ be constructed as (12). Denote by $L_{g_i^{(\ell)}}$ the Lipschitz constant of $g_i^{(\ell)} \in \mathcal{C}_{\mathrm{Lip}}(\mathbb{R}^{d_\ell}, \mathbb{R}^{d_{\ell+1}})$. Let $\varepsilon > 0$.

We build the following continuous GNN: $\bar{f}^{(0)} = f^{(0)}$, and

$$\bar{f}^{(\ell+1)} = f_{\theta_1^{(\ell)}}^{\mathrm{MLP}} \circ \bar{f}^{(\ell)} + f_{\theta_2^{(\ell)}}^{\mathrm{MLP}} \circ \mathbf{S}\bar{f}^{(\ell)}$$

$$\Phi_\theta(\mathbf{S}, \bar{f}^{(0)}) = \bar{f}^{(L)}$$

for well-chosen MLPs. We design them by increasing layer indices: assuming the MLPs up to layer $\ell - 1$ are choosen (i.e. $\bar{f}^{(\ell)}$ is chosen), we use Lemma 3 and choose $\theta_i^{(\ell)}$ (which depends on $\theta^{(0)}, \ldots, \theta^{(\ell-1)}$ then) such that

$$\left\| \left( g_1^{(\ell)} - f_{\theta_1^{(\ell)}}^{\mathrm{MLP}} \right) \circ \bar{f}^{(\ell)} \right\|_{L^2} + \left\| \left( g_2^{(\ell)} - f_{\theta_2^{(\ell)}}^{\mathrm{MLP}} \right) \circ \mathbf{S}\bar{f}^{(\ell)} \right\|_{L^2} \leqslant \varepsilon^{(\ell)} \overset{\text{def.}}{=} \frac{\varepsilon}{L \prod_{q=\ell+1}^{L-1} \left( L_{g_1^{(\ell)}} + L_{g_2^{(\ell)}} \|\mathbf{S}\| \right)}$$

Then we get

$$\left\| f^{(\ell+1)} - \bar{f}^{(\ell+1)} \right\|_{L^2} \leqslant \left\| g_1^{(\ell)} \circ f^{(\ell)} - f_{\theta_1^{(\ell)}}^{\mathrm{MLP}} \circ \bar{f}^{(\ell)} \right\|_{L^2} + \left\| g_2^{(\ell)} \circ \mathbf{S} f^{(\ell)} - f_{\theta_2^{(\ell)}}^{\mathrm{MLP}} \circ \mathbf{S} \bar{f}^{(\ell)} \right\|_{L^2}$$

$$\leqslant \left\| g_1^{(\ell)} \circ f^{(\ell)} - g_1^{(\ell)} \circ \bar{f}^{(\ell)} \right\|_{L^2} + \left\| g_2^{(\ell)} \circ \mathbf{S} f^{(\ell)} - g_2^{(\ell)} \circ \mathbf{S} \bar{f}^{(\ell)} \right\|_{L^2} + \varepsilon^{(\ell)}$$

$$\leqslant \left( L_{g_1^{(\ell)}} + L_{g_2^{(\ell)}} \|\mathbf{S}\| \right) \left\| f^{(\ell)} - \bar{f}^{(\ell)} \right\|_{L^2} + \varepsilon^{(\ell)}$$

Hence by a simple recursion and since $f^{(0)} = \bar{f}^{(0)}$ we have

$$\left\| f^{(L)} - \bar{f}^{(L)} \right\|_{L^2} \leqslant \sum_{\ell=0}^{L-1} \left( \prod_{q=\ell+1}^{L-1} \left( L_{g_1^{(\ell)}} + L_{g_2^{(\ell)}} \|\mathbf{S}\| \right) \right) \varepsilon^{(\ell)} \leqslant \varepsilon$$

by our choice of $\varepsilon^{(\ell)}$, which concludes the proof. $\qquad\square$

*Proof of Theorem 1.* We start with the inclusion $\mathcal{F}_{\mathbf{S}} \subseteq \mathcal{F}_{\mathrm{GNN}}$. Let $f \in \mathcal{F}_{\mathbf{S}}(\mathcal{B})$ and $\varepsilon > 0$. By Lemma 2, there is $f_c \in \mathcal{F}_c(\mathcal{B})$ constructed as (12) such that $\|f - f_c\|_{L^2} \leqslant \varepsilon/3$, and we use the weak law of large numbers to obtain that $\mathbb{P}(\|\iota_X(f - f_c)\|_{\mathrm{MSE}} \geqslant \varepsilon/3) \xrightarrow{n\to\infty} 0.$. By Lemma 4, there exists $\theta$ such that $\left\| \Phi_\theta(\mathbf{S}, f^{(0)}) - f_c \right\|_{L^2} \leqslant \varepsilon/3$. Again by the LLN, we have that $\mathbb{P}(\left\| \iota_X(\Phi_\theta(\mathbf{S}, f^{(0)}) - f_c) \right\|_{\mathrm{MSE}} \geqslant \varepsilon/3) \to 0$. Finally, by Lemma 1), we also have

$$\mathbb{P}\left( \left\| \Phi_\theta(S, \iota_X f^{(0)}) - \iota_X \Phi_\theta(\mathbf{S}, f^{(0)}) \right\|_{\mathrm{MSE}} \geqslant \varepsilon/3 \right) \xrightarrow{n\to\infty} 0.$$

Using a triangular inequality, we have

$$\left\| \Phi_\theta(\iota_X f^{(0)}) - \iota_X f \right\|_{\mathrm{MSE}} \leqslant \left\| \Phi_\theta(\iota_X f^{(0)}) - \iota_X \Phi_\theta(f^{(0)}) \right\|_{\mathrm{MSE}} + \left\| \iota_X(\Phi_\theta(f^{(0)}) - f_c) \right\|_{\mathrm{MSE}}$$
$$+ \|\iota_X(f_c - f)\|_{\mathrm{MSE}}.$$

We conclude by a union bound, and $f \in \mathcal{F}_{\mathrm{GNN}}(\mathcal{B})$.

For the reverse inclusion, let $f \in \mathcal{F}_{\mathrm{GNN}}(\mathcal{B})$. By hypothesis, for all $m \in \mathbb{N}$, there are $\theta \in \Theta$, $f^{(0)} \in \mathcal{B}$ such that

$$\mathbb{P}\left( \left\| \Phi_\theta(S, \iota_X f^{(0)}) - \iota_X f \right\|_{\mathrm{MSE}} \geqslant 1/m \right) \xrightarrow{n\to\infty} 0$$

By Lemma 1,

$$\mathbb{P}\left( \left\| \Phi_\theta(S, \iota_X f^{(0)}) - \iota_X \Phi_\theta(\mathbf{S}, f^{(0)}) \right\|_{\mathrm{MSE}} \geqslant 1/m \right) \xrightarrow{n\to\infty} 0$$

By the LLN,

$$\mathbb{P}\left( \left| \left\| \iota_X(f - \Phi_\theta(\mathbf{S}, f^{(0)})) \right\|_{\mathrm{MSE}} - \left\| f - \Phi_\theta(\mathbf{S}, f^{(0)}) \right\|_{L^2} \right| \geqslant 1/m \right) \to 0$$

Hence, b a union bound and triangular inequality, we obtain the deterministic bound $\left\| f - \Phi_\theta(\mathbf{S}, f^{(0)}) \right\|_{L^2} \leqslant 3/m$. Since $\Phi_\theta(\mathbf{S}, f^{(0)}) \in \mathcal{F}_{\mathbf{S}}(\mathcal{B})$ and $\mathcal{F}_{\mathbf{S}}(\mathbf{S})$ is closed, by taking $m \to \infty$ we have $f \in \mathcal{F}_{\mathbf{S}}(\mathcal{B})$. $\qquad\square$

## A.4  Proof of Prop. 3

Remark that $\mathcal{F}_{\mathbf{S}}(\mathcal{B}) \cap \mathcal{C}_{\mathrm{Lip}}(\mathcal{X}, \mathbb{R})$ is in fact a subalgebra of $\mathcal{C}_{\mathrm{Lip}}(\mathcal{X}, \mathbb{R})$. Indeed it is a vector space, and moreover it is stable by multiplication: for $f, g \in \mathcal{F}_{\mathbf{S}}(\mathcal{B}) \cap \mathcal{C}_{\mathrm{Lip}}(\mathcal{X}, \mathbb{R})$, by stability of $\mathcal{F}_{\mathbf{S}}(\mathcal{B})$ by composition with continuous functions we have that $x \mapsto [f(x), 0]$, $x \mapsto [0, g(x)]$ are in $\mathcal{F}_{\mathbf{S}}(\mathcal{B})$, then $(x \mapsto [f(x), g(x)]) \in \mathcal{F}_{\mathbf{S}}(\mathcal{B})$ by linearity (that is, $\mathcal{F}_{\mathbf{S}}$ is stable by concatenation), and since $(x, y) \mapsto xy$ is continuous, $(x \mapsto f(x)g(x)) \in \mathcal{F}_{\mathbf{S}}(\mathcal{B}) \cap \mathcal{C}_{\mathrm{Lip}}(\mathcal{X}, \mathbb{R})$.

Hence $\mathcal{F}_{\mathbf{S}}(\mathcal{B}) \cap \mathcal{C}_{\mathrm{Lip}}(\mathcal{X}, \mathbb{R})$ is a subalgebra of $\mathcal{C}_{\mathrm{Lip}}(\mathcal{X}, \mathbb{R})$ and since it separates points by hypothesis, by the Stone-Weierstrass theorem [21] it is dense in $\mathcal{C}_{\mathrm{Lip}}(\mathcal{X}, \mathbb{R})$ for the uniform norm, and *a fortiori* in $\sqcup_d \mathcal{C}_{\mathrm{Lip}}(\mathcal{X}, \mathbb{R}^d)$ by concatenation. Since continuous functions are dense in square-integrable functions [14, Sec. 8.2] and the $L^2$ norm is dominated by the uniform norm, it results that $\mathcal{F}_{\mathbf{S}}(\mathcal{B})$ is dense in $L^2_{\sqcup}$, and even $\mathcal{F}_{\mathbf{S}}(\mathcal{B}) = L^2_{\sqcup}$ because it is closed.

# B  Proof of Sec. 4

We introduce general notations that are valid all throughout this section of appendix. In this appendix, we will only assume:

**Assumption 2.** *We have the following.*

1. *a bounded kernel $w_{\mathbf{S}}$, and either $w_{\mathbf{S}}$ is p.s.d. or $\mathcal{X}$ is finite;*

2. *the operator $\mathbf{S}f = \int w_{\mathbf{S}}(\cdot, x)dP(x)$;*

3. *the Gram matrix $W = [w_{\mathbf{S}}(x_i, x_j)/n]$;*

4. *a graph matrix $S$ such that*
$$\|S - W\| \to 0 \tag{15}$$
*in probability.*

These assumptions are verified for **both** adjacency and normalized Laplacian: in the first case, we take $w_{\mathbf{S}} = w$, and $\|A/(\alpha_n n) - W\| \to 0$ by Theorem 9, and in the second, we take $w_{\mathbf{S}}(x, y) = \frac{w(x,y)}{\sqrt{d(x)d(y)}}$, which is bounded by our assumptions on $d$, and p.s.d. when $w$ is itself p.s.d., and we have indeed $\|L - W\| \to 0$ by Theorem 9. We will also use the following property.

**Lemma 5.** *$\mathcal{F}_{\mathrm{PE}}$ is closed.*

*Proof.* Let $f_m$ be a sequence in $\mathcal{F}_{\mathrm{PE}}$ that converges to a $f \in L_q^2$ for some $q$. Remark that $f_m \in L_q^2$ for all $m$ big enough. Let $\varepsilon > 0$, and $m$ be such that $\|f_m - f\|_{L^2} \leqslant \varepsilon/2$. By the law of large numbers, for any fixed $m$, $\|\iota_X(f_m - f)\|_{\mathrm{MSE}}$ converges to $\|f_m - f\|_{L^2} \leqslant \varepsilon/2$ almost surely and *a fortiori* in probability, such that $\mathbb{P}(\|\iota_X(f_m - f)\|_{\mathrm{MSE}} \geqslant \varepsilon/2) \xrightarrow[n \to \infty]{} 0$. By definition, there is $\gamma \in \Gamma$ such that $\mathbb{P}(\|\mathrm{PE}_\gamma(S, Z) - \iota_X f_m\|_{\mathrm{MSE}} \geqslant \varepsilon/2) \to 0$. Hence, by a union bound

$$\mathbb{P}(\|\mathrm{PE}_\gamma(S, Z) - \iota_X f\|_{\mathrm{MSE}} \geqslant \varepsilon)$$
$$\leqslant \mathbb{P}(\|\mathrm{PE}_\gamma(S, Z) - \iota_X f_m\|_{\mathrm{MSE}} \geqslant \varepsilon/2) + \mathbb{P}(\|\iota_X(f_m - f)\|_{\mathrm{MSE}} \geqslant \varepsilon/2) \xrightarrow[n \to \infty]{} 0$$

and therefore $f \in \mathcal{F}_{\mathrm{PE}}$. $\qquad\square$

## B.1  Proof of Prop. 4

We have

$$\left\| Z - \iota_X \mathbf{S} f^{(0)} \right\|_{\mathrm{MSE}} \leqslant \left\| S \iota_X f^{(0)} - \iota_X \mathbf{S} f^{(0)} \right\|_{\mathrm{MSE}} + \|S\nu\|_{\mathrm{MSE}}$$

The first term goes to 0 by Prop. 2.

For the second term, using $\|AB\|_{\mathrm{F}} \leqslant \|A\| \|B\|_{\mathrm{F}}$, we have

$$\|S\nu\|_{\mathrm{MSE}} \leqslant \|S - W\| \|\nu\|_{\mathrm{MSE}} + \|W\nu\|_{\mathrm{MSE}}$$

The first term goes to 0 in probability since $\|\varepsilon\|_{\mathrm{MSE}}$ is bounded with probability going to 1 and $\|S - W\| \to 0$ for our examples of graph operators. For the second term, we have

$$\|W\nu\|_{\mathrm{MSE}}^2 = \sum_{\ell=1}^{d_0} \frac{1}{n} \sum_i \left( \frac{1}{n} \sum_j w_{\mathbf{S}}(x_i, x_j)\nu_{j\ell} \right)^2 \leqslant \sum_\ell \left\| \frac{1}{n} \sum_j w_{\mathbf{S}}(\cdot, x_j)\nu_{j\ell} \right\|_\infty^2$$

Using Lemma 8 with the iid variable $y_j = (x_j, \nu_{j\ell})$ and $\mathbb{E}w_{\mathbf{S}}(\cdot, x_j)\nu_{j\ell} = 0$ since $\nu$ and $X$ are independent, we obtain

$$\forall \ell, \quad \left\| \frac{1}{n} \sum_j w_{\mathbf{S}}(\cdot, x_j)\nu_{j\ell} \right\|_\infty \xrightarrow[n \to \infty]{\mathcal{P}} 0$$

which concludes the proof.

## B.2 Eigenvectors positional encodings: SignNet

In this whole section, the number of eigenvectors $q$ is fixed, and we assume that $\lambda_1^{\mathbf{S}}, \ldots, \lambda_{q+1}^{\mathbf{S}}$ are pairwise distinct. We first start by generic results that allows to go from the graph matrix $S$ to the Gram matrix.

**Lemma 6** (Intermediate result for SignNet). *Suppose that Assumption 2 holds, that $\lambda_1^{\mathbf{S}}, \ldots, \lambda_{q+1}^{\mathbf{S}}$ are distinct, and that*

$$\max_{i=1,\ldots,q} \min_{s \in \{1,-1\}} \left\| s\sqrt{n}u_i^W - \iota_X u_i^{\mathbf{S}} \right\|_{\mathrm{MSE}} + \left| \lambda_i^W - \lambda_i^{\mathbf{S}} \right| \xrightarrow[n \to \infty]{\mathcal{P}} 0. \tag{16}$$

*Then the result of Theorem 2 holds.*

*Proof.* Let $f \in \mathcal{F}_{\mathrm{Eig}}$, written as (9). By Assumption 2, $\|S - W\| \to 0$. By Kato's inequality, we have that $\sup_i \left| \lambda_i^S - \lambda_i^W \right| \to 0$, and by hypothesis, the eigenvalues of $W$ converge to those of $\mathbf{S}$. Given the hypotheses on the eigenvalues of $\mathbf{S}$, with probability going to 1 the $q+1$ first eigenvalues of $S$ have single multiplicities. When it is the case, according to Davis-Kahan theorem (Theorem 8), for all $i = 1, \ldots, q$ there is $s_i \in \{-1, 1\}$ such that

$$\max_i \left\| s_i u_i^S - u_i^W \right\| \to 0 \tag{17}$$

which, combined with our hypotheses, yields

$$\max_{i=1,\ldots,q} \min_{s \in \{1,-1\}} \left\| s\sqrt{n}u_i^S - \iota_X u_i^{\mathbf{S}} \right\|_{\mathrm{MSE}} \xrightarrow[n \to \infty]{\mathcal{P}} 0. \tag{18}$$

For $i = 1, \ldots, q$, let $f_i : \mathbb{R} \to \mathbb{R}^{p_i}$ be continuous functions and $\varepsilon > 0$. By Lemma 3, there is $f_{\gamma_i}^{\mathrm{MLP}}$ such that $\left\| (f_{\gamma_i}^{\mathrm{MLP}} - f_i) \circ u_i^{\mathbf{S}} \right\|_{L^2} \leqslant \varepsilon/(2q)$. Then, call $L_i$ the (uniform) Lipschitz constant of $f_{\gamma_i}^{\mathrm{MLP}}$ on $\mathbb{R}$. We have

$$
\begin{aligned}
\left\| \mathrm{PE}_\gamma - \iota_X [(\mathbf{Q}f_i) \circ u_i^{\mathbf{S}}]_{i=1}^q \right\|_{\mathrm{MSE}} &= \left\| [(\mathbf{Q}f_{\gamma_i}^{\mathrm{MLP}})(\sqrt{n}u_i^S)]_{i=1}^q - \iota_X [(\mathbf{Q}f_i) \circ u_i^{\mathbf{S}}]_{i=1}^q \right\|_{\mathrm{MSE}} \\
&\leqslant \left\| [(\mathbf{Q}f_{\gamma_i}^{\mathrm{MLP}})(\sqrt{n}u_i^S)]_{i=1}^q - \iota_X [(\mathbf{Q}f_{\gamma_i}^{\mathrm{MLP}}) \circ u_i^{\mathbf{S}}]_{i=1}^q \right\|_{\mathrm{MSE}} \\
&\quad + \left\| \iota_X [(\mathbf{Q}f_{\gamma_i}^{\mathrm{MLP}}) \circ u_i^{\mathbf{S}}]_{i=1}^q - \iota_X [(\mathbf{Q}f_i) \circ u_i^{\mathbf{S}}]_{i=1}^q \right\|_{\mathrm{MSE}} \\
&\leqslant \sum_i \left\| (\mathbf{Q}f_{\gamma_i}^{\mathrm{MLP}})(\sqrt{n}u_i^S) - \iota_X (\mathbf{Q}f_{\gamma_i}^{\mathrm{MLP}}) \circ u_i^{\mathbf{S}} \right\|_{\mathrm{MSE}} \\
&\quad + \left\| \iota_X (\mathbf{Q}f_{\gamma_i}^{\mathrm{MLP}}) \circ u_i^{\mathbf{S}} - \iota_X (\mathbf{Q}f_i) \circ u_i^{\mathbf{S}} \right\|_{\mathrm{MSE}} \\
&\leqslant 2 \sum_i \min_{s_i \in \{1,-1\}} \left\| f_{\gamma_i}^{\mathrm{MLP}}(s_i\sqrt{n}u_i^S) - \iota_X f_{\gamma_i}^{\mathrm{MLP}} \circ u_i^{\mathbf{S}} \right\|_{\mathrm{MSE}} \\
&\quad + 2 \left\| \iota_X (f_{\gamma_i}^{\mathrm{MLP}} - f_i) \circ u_i^{\mathbf{S}} \right\|_{\mathrm{MSE}}
\end{aligned}
$$

The first term goes to 0 in probability by what precedes, while for the second

$$\sum_i \left\| \iota_X (f_{\gamma_i}^{\mathrm{MLP}} - f_i) \circ u_i^{\mathbf{S}} \right\|_{\mathrm{MSE}} \xrightarrow[n \to \infty]{\mathcal{P}} \sum_i \left\| (f_{\gamma_i}^{\mathrm{MLP}} - f_i) \circ u_i^{\mathbf{S}} \right\|_{L^2} \leqslant \varepsilon/2$$

which proves that $\mathcal{F}_{\mathrm{Eig}} \subset \mathcal{F}_{\mathrm{PE}}$, and thus $\overline{\mathcal{F}_{\mathrm{Eig}}} \subset \mathcal{F}_{\mathrm{PE}}$ since $\mathcal{F}_{\mathrm{PE}}$ is closed by Lemma 5.

For the reverse inclusion, let $f \in \mathcal{F}_{\mathrm{PE}}$. By hypothesis, for all $m \in \mathbb{N}$, there is $\gamma \in \Gamma$, such that

$$\mathbb{P}\left( \left\| \mathrm{PE}_\gamma - \iota_X f \right\|_{\mathrm{MSE}} \geqslant 1/m \right) \xrightarrow[n \to \infty]{} 0$$

By what precedes, if we write $\mathrm{PE}_\gamma = (\mathbf{Q}f_{\gamma_i}^{\mathrm{MLP}})(\sqrt{n}u_i^S)]_{i=1}^q$, we have

$$\left\| \mathrm{PE}_\gamma - \iota_X [(\mathbf{Q}f_{\gamma_i}^{\mathrm{MLP}}) \circ u_i^{\mathbf{S}}]_{i=1}^q \right\|_{\mathrm{MSE}} \xrightarrow[n \to \infty]{\mathcal{P}} 0$$

By the LLN,

$$\mathbb{P}\left( \left| \left\| \iota_X (f - [(\mathbf{Q}f_{\gamma_i}^{\mathrm{MLP}}) \circ u_i^{\mathbf{S}}]_{i=1}^q) \right\|_{\mathrm{MSE}} - \left\| f - [(\mathbf{Q}f_{\gamma_i}^{\mathrm{MLP}}) \circ u_i^{\mathbf{S}}]_{i=1}^q \right\|_{L^2} \right| \geqslant 1/m \right) \to 0$$

Hence, by a union bound and triangular inequality, we obtain the deterministic bound $\left\| f - [(\mathbf{Q}f_{\gamma_i}^{\mathrm{MLP}}) \circ u_i^{\mathbf{S}}]_{i=1}^q \right\|_{L^2} \leqslant 3/m$. Since $[(\mathbf{Q}f_{\gamma_i}^{\mathrm{MLP}}) \circ u_i^{\mathbf{S}}]_{i=1}^q \in \mathcal{F}_{\mathrm{Eig}}$, we have $f \in \overline{\mathcal{F}_{\mathrm{Eig}}}$. $\square$

We must now prove the hypothesis of Lemma 6. This is done separately for p.s.d. kernel and SBM.

### B.2.1 Positive semidefinite kernels

In this subsection, we assume that $w_\mathbf{S}$ is p.s.d. The following result is adapted from [42].

**Theorem 5** (Adapted from [42]). *Suppose that Assumption 2 holds, that $\lambda_1^\mathbf{S}, \ldots, \lambda_{q+1}^\mathbf{S}$ are pairwise distinct, and that $w_\mathbf{S}$ is p.s.d. (Ex. b). Then:*

$$\max_{i=1,\ldots,q} \min_{s \in \{1,-1\}} \left\| s\sqrt{n}u_i^W - \iota_X u_i^\mathbf{S} \right\|_{\mathrm{MSE}} + \left| \lambda_i^W - \lambda_i^\mathbf{S} \right| \to 0 \tag{19}$$

*in probability.*

*Proof.* Denote by $\mathcal{H}$ the RKHS associated with $w_\mathbf{S}$, and by $T_\mathcal{H} : \mathcal{H} \to \mathcal{H}$ the kernel integral operator. Then, it is known [42] that the spectrum of $\mathbf{S}$ and $T_\mathcal{H}$ are the same (up to $0$'s), and the eigenfunctions (normalized in $\mathcal{H}$) $v_i$ of $T_\mathcal{H}$ corresponding to positive eigenvalues satisfy:

$$v_i^\mathbf{S}(x) = \begin{cases} \sqrt{\lambda_i^\mathbf{S}} u_i^\mathbf{S}(x) & \text{for } x \in supp(P) \\ \frac{1}{\sqrt{\lambda_i^\mathbf{S}}} \int w_\mathbf{S}(x,y) u_i^\mathbf{S}(y) dP(y) & \text{else} \end{cases} \tag{20}$$

Note that $\lambda_1^\mathbf{S} > \ldots > \lambda_q^\mathbf{S} > \lambda_{q+1}^\mathbf{S} \geqslant 0$ by hypothesis. Following [42], we know that

$$\sup_i \left| \lambda_i^W - \lambda_i^\mathbf{S} \right| \to 0 \tag{21}$$

in probability, and that in particular, with probability going to one $\lambda_i^W > 0$ for all $i = 1, \ldots, q$. Assuming this is satisfied for all $i$, we denote by $v_i^W = \frac{1}{\sqrt{n\lambda_i^W}} \sum_j w_\mathbf{S}(\cdot, x_j) u_{i,j}^W \in \mathcal{H}$. Then, using the fact that the eigenvalues of $\mathbf{S}$ have single multiplicities, the proof of Theorem 12 in [42] tells us that for all $m = 1, \ldots, q$,

$$\sum_{j=1}^m \sum_{i \geqslant m+1} \left\langle v_i^\mathbf{S}, v_j^W \right\rangle_\mathcal{H}^2 + \sum_{j \geqslant m+1} \sum_{i=1}^m \left\langle v_i^\mathbf{S}, v_j^W \right\rangle_\mathcal{H}^2 \to 0 \tag{22}$$

in probability. Since these are all nonnegative quantities, all partial sums go to $0$. The first part (the term for $j = m$) tells us that $\sum_{i \geqslant m+1} \left\langle v_i^\mathbf{S}, v_m^W \right\rangle_\mathcal{H}^2 \to 0$, and the second part applied at $m-1$ (again the term with $j = m$) gives us $\sum_{i=1}^{m-1} \left\langle v_i^\mathbf{S}, v_m^W \right\rangle_\mathcal{H}^2 \to 0$. Hence for all $m = 1, \ldots, q$,

$$\sum_{i \neq m} \left\langle v_i^\mathbf{S}, v_m^W \right\rangle_\mathcal{H}^2 \to 0 \tag{23}$$

Since $(v_i^\mathbf{S})_i$ and $(v_i^W)_i$ are orthonormal basis of $\mathcal{H}$, we have $\left\| v_m^W \right\|_\mathcal{H}^2 = 1 = \sum_i \left\langle v_i^\mathbf{S}, v_m^W \right\rangle_\mathcal{H}^2$, and thus $\left\langle v_m^\mathbf{S}, v_m^W \right\rangle_\mathcal{H}^2 \to 1$. By the reproducing property

$$\left\langle v_m^\mathbf{S}, v_m^W \right\rangle_\mathcal{H} = \frac{1}{\sqrt{\lambda_m^W}} \frac{1}{\sqrt{n}} \sum_i u_{m,i}^W v_m^\mathbf{S}(x_i) = \sqrt{\frac{\lambda_m^\mathbf{S}}{\lambda_m^W}} \frac{1}{\sqrt{n}} \sum_i u_{m,i}^W u_m^\mathbf{S}(x_i)$$

By the convergence of $\lambda_m^W$ we obtain that $\left( \frac{1}{\sqrt{n}} \left\langle u_i^W, \iota_X u_i^\mathbf{S} \right\rangle \right)^2 \to 1$ in probability, and choosing the sign of $u_i^W$ such that $\left\langle u_i^W, \iota_X u_i^\mathbf{S} \right\rangle \geqslant 0$, we get $\frac{1}{\sqrt{n}} \left\langle u_i^W, \iota_X u_i^\mathbf{S} \right\rangle \to 1$. Finally

$$\left\| \sqrt{n}u_i^W - \iota_X u_i^\mathbf{S} \right\|_{\mathrm{MSE}}^2 = 2\left( 1 - \frac{1}{\sqrt{n}} \left\langle u_i^W, \iota_X u_i^\mathbf{S} \right\rangle \right) + \left\| \iota_X u_i^\mathbf{S} \right\|_{\mathrm{MSE}}^2 - 1$$

by the LLN, $\left\| \iota_X u_i^\mathbf{S} \right\|_{\mathrm{MSE}}^2 \to \left\| u_i^\mathbf{S} \right\|_{L^2}^2 = 1$ in probability, which concludes the proof. $\square$

By combining Theorem 5 with Lemma 6, we conclude the proof in the p.s.d. case.

### B.2.2 SBM

Variants of the following result appear under (sometimes significantly) different formulations in the literature.

**Theorem 6.** *Suppose that Assumption 2 holds, that $\lambda_1^{\mathbf{S}}, \dots, \lambda_{q+1}^{\mathbf{S}}$ are pairwise distinct, and that $\mathcal{X}$ is finite (Ex. a). Then:*

$$\max_{i=1,\dots,q} \min_{s \in \{1,-1\}} \left\| s\sqrt{n} u_i^W - \iota_X u_i^{\mathbf{S}} \right\|_{\mathrm{MSE}} + \left| \lambda_i^W - \lambda_i^{\mathbf{S}} \right| \xrightarrow[n\to\infty]{\mathcal{P}} 0. \tag{24}$$

*Proof.* In the SBM case, functions are represented by vectors of size $K$. The operator $\mathbf{S}$ acts as: $\mathbf{S}f = C\operatorname{diag}(P_k)f$, where $C \stackrel{\text{def.}}{=} [w_{\mathbf{S}}(k,\ell)]_{k\ell}$. Therefore $C\operatorname{diag}(P_k)u_i^{\mathbf{S}} = \lambda_i^{\mathbf{S}} u_i^{\mathbf{S}}$, for $i = 1, \dots, K$. Note that $u_i^{\mathbf{S}}$ is orthonormal *in* $L^2(P)$, that is, $(u_i^{\mathbf{S}})^\top \operatorname{diag}(P_k) u_i^{\mathbf{S}} = 1$ and $(u_i^{\mathbf{S}})^\top \operatorname{diag}(P_k) u_j^{\mathbf{S}} = 0$. In particular, $(\lambda_i^{\mathbf{S}}, \operatorname{diag}(\sqrt{P_k})u_i^{\mathbf{S}})$ is an eigenvalue/eigenvector pair of the symmetric matrix $C_P \stackrel{\text{def.}}{=} \operatorname{diag}(\sqrt{P_k})C\operatorname{diag}(\sqrt{P_k})$. Denoting by $u_i^P \stackrel{\text{def.}}{=} \operatorname{diag}(\sqrt{P_k})u_i^{\mathbf{S}}$ we have $C_P = \sum_{i=1}^K \lambda_i^{\mathbf{S}} u_i^P (u_i^P)^\top$.

Since the space $\mathcal{X}$ is finite, each $x_i$ is equal to a community label $1 \leqslant k_i \leqslant K$. Define $\Theta \in \{0, 1/\sqrt{n}\}^{n \times K}$ the *community matrix*, as:

$$\begin{cases} \Theta_{i,k_i} = \frac{1}{\sqrt{n}} & \text{for all } 1 \leqslant i \leqslant n \\ \Theta_{i,j} = 0 & \text{otherwise} \end{cases} \tag{25}$$

Then the Gram matrix is

$$W = \Theta C \Theta^\top \tag{26}$$

Also note that $\Theta^\top \Theta = \operatorname{diag}(\hat{P})$, where $\hat{P}_k = \frac{1}{n}\sum_i 1_{k_i=k} \to P_k$ almost surely by the LLN. Note that, when interpreting $u_i^{\mathbf{S}}$ as a function on $\mathcal{X} = \{1, \dots, K\}$, we have $\sqrt{n}\Theta u_i^{\mathbf{S}} = \iota_X u_i^{\mathbf{S}}$.

Defining $\Theta_P = \Theta\operatorname{diag}(1/\sqrt{P_k})$, we have

$$W = \Theta_P C_P \Theta_P^\top = \sum_i \lambda_i^{\mathbf{S}} (\Theta_P u_i^P)(\Theta_P u_i^P)^\top = \sum_i \lambda_i^{\mathbf{S}} v_i v_i^\top$$

where $v_i = \Theta_P u_i^P = \Theta u_i^{\mathbf{S}}$ satisfies

$$\|v_i\|^2 = (u_i^{\mathbf{S}})^\top \operatorname{diag}(\hat{P}_k) u_i^{\mathbf{S}} \to 1, \quad v_i^\top v_j = (u_i^{\mathbf{S}})^\top \operatorname{diag}(\hat{P}_k) u_j^{\mathbf{S}} \to 0 \tag{27}$$

in probability. Hence the $v_i$ are almost orthonormal but not exactly. Define their orthonormalization:

$$\tilde{u}_1 = v_1, \ \forall i = 2, \dots, K, \ \tilde{u}_i = v_i - \sum_{j=1}^{i-1}(v_i^\top u_j)u_j \text{ and } u_i = \frac{\tilde{u}_i}{\|\tilde{u}_i\|} \tag{28}$$

such that $u_1, \dots, u_K \in \mathbb{R}^n$ are orthonormal and $u_i \in \operatorname{Span}(v_1, \dots, v_i)$. We define $G = \sum_{i=1}^K \lambda_i^{\mathbf{S}} u_i u_i^\top$, whose eigenvalues are the $\lambda_i^{\mathbf{S}}$ and eigenvectors $u_i$. By the properties of $v_i$ we have $\sup_{1 \leqslant i \leqslant n} \|v_i - u_i\| \to 0$, so $\|W - G\| \to 0$. Hence by Kato's inequality we have $\sup_{1 \leqslant i \leqslant n} |\lambda_i^W - \lambda_i^{\mathbf{S}}| \to 0$, and since the $\lambda_1^{\mathbf{S}}, \dots, \lambda_{q+1}^{\mathbf{S}}$ have unique multiplicity, again by Davis-Kahan theorem for all $i = 1, \dots, q$ we have $\min_{s \in \{-1,1\}} \|su_i^W - u_i\| \to 0$, and by consequence $\min_{s \in \{-1,1\}} \|s\sqrt{n}u_i^W - \sqrt{n}v_i\|_{\mathrm{MSE}} \to 0$. We conclude by recalling that $\sqrt{n}v_i = \sqrt{n}\Theta u_i^{\mathbf{S}} = \iota_X u_i^{\mathbf{S}}$. $\square$

As before, we conclude by combining Theorem 6 with Lemma 6.

## B.3 Distance-based encodings

Again, we start with an intermediate result, assuming some convergence in Frobenius norm that we will then show for our cases of interest.

**Lemma 7** (Intermediate result for Distance-based encodings). *Suppose that Assumption 2 holds, and that:*

$$\forall \varepsilon > 0, \exists \gamma_2, \mathbb{P}(\|S_{\gamma_2} - W\|_{\mathrm{F}} \geqslant \varepsilon) \to 0 \tag{29}$$

*Then the result of Theorem 3 holds.*

*Proof.* Let $f \in \mathcal{C}_{\mathrm{Lip}}([0,1]^q, \mathbb{R}^d)$ and $\varepsilon > 0$. By the universality theorem [40], let $f_{\gamma_1}^{\mathrm{MLP}}$ such that $\left\| f_{\gamma_1}^{\mathrm{MLP}} - f \right\|_\infty \leqslant \varepsilon$ on $[0,1]^q$. Since it is an MLP, $f_{\gamma_1}^{\mathrm{MLP}}$ is uniformly Lipschitz on $\mathbb{R}^q$, call $L$ its Lipschitz constant. Then, let $f_{\gamma_2}^{\mathrm{MLP}}$ such that

$$\mathbb{P}(\| S_{\lambda_2} - W \|_{\mathrm{F}} \geqslant \varepsilon/L) \to 0 \tag{30}$$

For convenience, define $M_j^{S_{\gamma_2}} = [S_{\gamma_2} e_j, \ldots, (S_{\gamma_2})^q e_j] \in \mathbb{R}^{n \times q}$, $M_j^W = [W e_j, \ldots, W^q e_j]$, $J(x,y) = [\mathbf{S} \delta_y(x), \ldots, \mathbf{S}^q \delta_y(x)] \in [0,1]^q$.

We write

$$
\begin{aligned}
& \left\| \mathrm{PE} - \iota_X \int f(J(\cdot, x)) dP(x) \right\|_{\mathrm{MSE}} \\
& \leqslant \left\| \frac{1}{n} \sum_j f_{\gamma_1}^{\mathrm{MLP}} \left( n \cdot M_j^{S_{\gamma_2}} \right) - f_{\gamma_1}^{\mathrm{MLP}} \left( n \cdot M_j^W \right) \right\|_{\mathrm{MSE}} \\
& \quad + \left\| \frac{1}{n} \sum_j f_{\gamma_1}^{\mathrm{MLP}} \left( n \cdot M_j^W \right) - \iota_X \int f_{\gamma_1}^{\mathrm{MLP}}(J(\cdot, x)) dP(x) \right\|_{\mathrm{MSE}} \\
& \quad + \left\| \iota_X \int f_{\gamma_1}^{\mathrm{MLP}}(J(\cdot, x)) dP(x) - \iota_X \int f(J(\cdot, x)) dP(x) \right\|_{\mathrm{MSE}} \tag{31}
\end{aligned}
$$

The third term in (31) is bounded by $\left\| f_{\gamma_1}^{\mathrm{MLP}} - f \right\|_\infty \leqslant \varepsilon$.

The second term in (31) is

$$
\begin{aligned}
& \left\| \frac{1}{n} \sum_j f_{\gamma_1}^{\mathrm{MLP}} \left( n \cdot M_j^W \right) - \iota_X \int f_{\gamma_1}^{\mathrm{MLP}}(J(\cdot, y)) dP(x) \right\|_{\mathrm{MSE}} \\
& \leqslant \left\| \frac{1}{n} \sum_j f_{\gamma_1}^{\mathrm{MLP}} \left( n \cdot M_j^W \right) - \iota_X f_{\gamma_1}^{\mathrm{MLP}}(J(\cdot, x_j)) \right\|_{\mathrm{MSE}} \\
& \quad + \left\| \iota_X \left( \frac{1}{n} \sum_j f_{\gamma_1}^{\mathrm{MLP}}(J(\cdot, x_j)) - \int f_{\gamma_1}^{\mathrm{MLP}}(J(\cdot, x)) dP(x) \right) \right\|_{\mathrm{MSE}} \\
& \leqslant L \sum_{\ell=1}^q \sup_{x, x'} \left| \frac{1}{n^{\ell-1}} \sum_{i_1, \ldots, i_{\ell-1}} w_{\mathbf{S}}(x, x_{i_1}) w_{\mathbf{S}}(x_{i_1}, x_{i_2}) \ldots w_{\mathbf{S}}(x_{i_{\ell-1}}, x') - \mathbf{S}^\ell \delta_{x'}(x) \right| \\
& \quad + \left\| \frac{1}{n} \sum_j f_{\gamma_1}^{\mathrm{MLP}}(J(\cdot, x_j)) - \int f_{\gamma_1}^{\mathrm{MLP}}(J(\cdot, x)) dP(x) \right\|_\infty
\end{aligned}
$$

where we have used the Lipschitz property of $f_{\gamma_1}^{\mathrm{MLP}}$ and a supremum over $x_i, x_j$ for the first term. The first term converges to 0 in probability by Lemma 9, while the second goes to 0, using the boundedness of $f_{\gamma_1}^{\mathrm{MLP}}$ on $[0,1]^q$, by Lemma 8.

Finally, using Schwartz inequality, the first term in (31) is bounded by

$$\left\| n^{-1} \sum_j f_{\gamma_1}^{\mathrm{MLP}}(nM_j^{S_{\gamma_2}}) - f_{\gamma_1}^{\mathrm{MLP}}(nM_j^W) \right\|_{\mathrm{MSE}}$$

$$\leqslant \frac{1}{\sqrt{n}} \sqrt{\sum_j \left\| f_{\gamma_1}^{\mathrm{MLP}}(nM_j^{S_{\gamma_2}}) - f_{\gamma_1}^{\mathrm{MLP}}(nM_j^W) \right\|_{\mathrm{MSE}}^2}$$

$$= \frac{1}{n} \sqrt{\sum_{ij} \left\| f_{\gamma_1}^{\mathrm{MLP}}(n(M_j^{S_{\gamma_2}})_{i,:}) - f_{\gamma_1}^{\mathrm{MLP}}(n(M_j^W)_{i,:}) \right\|^2}$$

$$\leqslant L \sqrt{\sum_{ij} \left\| (M_j^{S_{\gamma_2}})_{i,:} - (M_j^W)_{i,:} \right\|^2}$$

$$\leqslant L \sum_\ell \left\| S_{\gamma_2}^\ell - W^\ell \right\|_{\mathrm{F}} \leqslant L \sum_\ell \left( \sum_{p=1}^{\ell-1} \|S_{\gamma_2}\|_{\mathrm{F}}^p \|W\|_{\mathrm{F}}^{\ell-1-p} \right) \|S_{\gamma_2} - W\|_{\mathrm{F}} \lesssim q^2 \varepsilon$$

with probability going to 1, using (30) and the fact that $\|W\|_{\mathrm{F}}$ is bounded, and thus $\|S_{\gamma_2}\|_{\mathrm{F}}$ as well with probability going to 1. Gathering everything, $f \in \mathcal{F}_{\mathrm{PE}}$, which concludes the proof. □

We then prove this property for both p.s.d. kernel and SBM. The following Theorem is similar to Theorem 4, but under Assumption 2, which is true for ex. 1 and 2.

**Theorem 7** (Theorem 4 reformulated). *Suppose that Assumption 2 holds. For both p.s.d. kernel (Ex. b) or finite $\mathcal{X}$ (Ex. a), for all $\varepsilon > 0$, there is an MLP filter $S_\gamma = h_{f_\gamma^{\mathrm{MLP}}}(S)$ such that*

$$\forall \varepsilon > 0, \ \exists \gamma, \ \mathbb{P}(\|S_\gamma - W\|_{\mathrm{F}} \geqslant \varepsilon) \to 0 \tag{32}$$

*Proof.* Note that $\mathbf{S}$ is trace-class (both if $w$ is p.s.d. or in the SBM case), such that $\sum_i |\lambda_i^{\mathbf{S}}| < \infty$. In particular, $|\lambda_i^{\mathbf{S}}| = o(1/\sqrt{i})$. In the p.s.d. case, all $\lambda_i^{\mathbf{S}}$ are nonnegative. We define $\lambda_\varepsilon > 0$ and the support $T = \{i \mid |\lambda_i^{\mathbf{S}}| \geqslant \lambda_\varepsilon\}$

i) $\sum_{i \in T^c} (\lambda_i^{\mathbf{S}})^2 \leqslant \varepsilon/2$, which can be satisfied since $\mathbf{S}$ is trace class and therefore Hilbert-Schmidt

ii) $\sqrt{2|T|} \sup_{i \in T^c} |\lambda_i^{\mathbf{S}}| \leqslant \varepsilon/4$, which can be satisfied since $|\lambda_i^{\mathbf{S}}| = o(1/\sqrt{i})$

iii) $\inf_{i \in T, j \in T^c} |\lambda_i^{\mathbf{S}} - \lambda_j^{\mathbf{S}}| > 0$, which can be satisfied by choosing $\lambda_\varepsilon$ in a gap in the spectrum of $\mathbf{S}$, since all eigenvalues but 0 have finite multiplicities.

We will first start by approximating $W$ with an ideal filtered matrix $\hat{S} = \sum_{i \in T} \lambda_i^S u_i^S (u_i^S)^\top$. We define $G = \sum_{i \in T} \lambda_i^W u_i^W (u_i^W)^\top$. We decompose

$$\left\| \hat{S} - W \right\|_{\mathrm{F}} \leqslant \left\| \hat{S} - G \right\|_{\mathrm{F}} + \|G - W\|_{\mathrm{F}}$$

For the first term, since this matrix is of rank at most $2|T|$, we have

$$\left\| \hat{S} - G \right\|_{\mathrm{F}} \leqslant \sqrt{2|T|} \left\| \hat{S} - G \right\|$$

We further decompose

$$\left\| \hat{S} - G \right\| \leqslant \left\| \hat{S} - S \right\| + \|S - W\| + \|W - G\|$$

The second term is bounded by Theorem 9, we have $\|S - W\| \to 0$ in probability.

The third term is bounded by Kato inequality

$$\sup_{i \in T^c} \lambda_i^W \leqslant \sup_{i \in T^c} \lambda_i^{\mathbf{S}} + \sup_i |\lambda_i^W - \lambda_i^{\mathbf{S}}|$$

the last term goes to 0 in probability.

The first term is bounded by $\sup_{i \in T^c} \lambda_i^S$, and by Kato's inequality

$$\sup_{i \in T^c} \lambda_i^S \leqslant \sup_{i \in T^c} \lambda_i^W + \|S - W\|$$

which is a combination of both previous cases.

At the end of the day, with probability going to 1,

$$\left\|\hat{S} - G\right\| \leqslant \sqrt{2\,|T|}\, 2 \sup_{i \in T^c} \lambda_i^{\mathbf{S}} + o(1) \leqslant \varepsilon/2 + o(1)$$

Finally, we have

$$\|G - W\|_{\mathrm{F}}^2 = \sum_{i \in T^c} (\lambda_i^W)^2 \leqslant 2 \left( \sum_{i \in T^c} (\lambda_i^{\mathbf{S}})^2 + \sum_i (\lambda_i^W - \lambda_i^{\mathbf{S}})^2 \right)$$

The first term is bounded by $\varepsilon/2$, the second goes to 0 in probability: in the p.s.d. case, this is a result of [42], by an application of Hoeffding's inequality in the Hilbert space of Hilbert-Schmidt operators in an RKHS; in the SBM case, there is a finite number of non-zero eigenvalues for both $W$ and $\mathbf{S}$ that converge to each other and the result follows.

Now we need to bound $\left\|S_\gamma - \hat{S}\right\|_{\mathrm{F}}$ for a well chosen MLP. Define $(i_T, j_T) = \arg\min_{i \in T, j \in T^c} \left|\lambda_i^{\mathbf{S}} - \lambda_j^{\mathbf{S}}\right|$, $\tau = \frac{\left|\lambda_{i_T}^{\mathbf{S}} - \lambda_{j_T}^{\mathbf{S}}\right|}{4} > 0$ and $\bar{\lambda} = \frac{\left|\lambda_{i_T}^{\mathbf{S}} + \lambda_{j_T}^{\mathbf{S}}\right|}{2}$. Note that we have $T = \{i \mid \left|\lambda_i^{\mathbf{S}}\right| \geqslant \bar{\lambda} + 2\tau\}$ and $T^c = \{i \mid \left|\lambda_i^{\mathbf{S}}\right| \leqslant \bar{\lambda} - 2\tau\}$ Define the following $f^{\mathrm{MLP}} : \mathbb{R} \to \mathbb{R}$:

$$f^{\mathrm{MLP}}(\lambda) = \frac{\bar{\lambda} + \tau}{2\tau} \left( \rho\left(\lambda - \bar{\lambda} + \tau\right) - \rho\left(\lambda - \bar{\lambda} - \tau\right) \right) + \rho\left(\lambda - \bar{\lambda} - \tau\right) \tag{33}$$

$$+ \frac{-\bar{\lambda} - \tau}{2\tau} \left( \rho\left(-\lambda - \bar{\lambda} + \tau\right) - \rho\left(-\lambda - \bar{\lambda} - \tau\right) \right) - \rho\left(-\lambda - \bar{\lambda} - \tau\right) \tag{34}$$

where $\rho$ is ReLU. This is a continuous piecewise linear function that is equal to $\lambda$ on $(-\infty, -\bar{\lambda} - \tau]$, 0 on $[-\bar{\lambda} + \tau, \bar{\lambda} - \tau]$, and $\lambda$ on $[\bar{\lambda} + \tau, +\infty)$.

Recall that with probability going to 1, we have $\sup_i \left|\lambda_i^S - \lambda_i^{\mathbf{S}}\right| \leqslant \tau$, so in particular, for all $i \in T$ we have $\left|\lambda_i^S\right| \geqslant \bar{\lambda} + \tau$ and for all $i \in T^c$ we have $\left|\lambda_i^S\right| \leqslant \bar{\lambda} - \tau$. In that case, the MLP filtering is exactly the ideal filtering and $h(S) = \hat{S}$, which concludes the proof. $\qquad\square$

### B.4  Proof of Prop. 5

The case of $\mathcal{F}_{\mathrm{Dist}}$ was proven in [25, Theorem 4]. For $\mathcal{F}_{\mathrm{Eig}}$, we consider the SBM (ex. a) with adjacency matrix (ex. 1) and

$$C = \begin{pmatrix} 1/2 & 1/4 \\ 1/4 & 3/8 \end{pmatrix}, \quad P = (1/3, 2/3)$$

We have $\mathbf{S}1 = (1/3, 1/3)$, therefore $\mathcal{F}_{\mathbf{S}}(1)$ contains only constant functions. Moreover, $u_i^{\mathbf{S}} = (1, 1)/\sqrt{2}$. From the orthogonality equation $(u_2^{\mathbf{S}})^\top \mathrm{diag}(P) u_1^{\mathbf{S}} = 0$ we get $(u_2^{\mathbf{S}})_1 = 2(u_2^{\mathbf{S}})_2$. Hence it is possible to choose $f$ such that $f((u_2^{\mathbf{S}})_1) + f(-(u_2^{\mathbf{S}})_1) \neq f((u_2^{\mathbf{S}})_2) + f(-(u_2^{\mathbf{S}})_2)$, thus $\mathcal{F}_{\mathrm{Eig}}$ contains a non-constant function, which concludes the proof.

### B.5  Proof of Prop. 6

Consider the SBM case $\mathcal{X} = \{1, \ldots, K\}$ with $K$ even and $P = 1_K/K$. Adopting the notations of the section above, we have $\mathbf{S}f = C\,\mathrm{diag}(P_k)f$, $C_P = \mathrm{diag}(\sqrt{P_k})C\,\mathrm{diag}(\sqrt{P_k}) = \sum_{i=1}^K \lambda_i^{\mathbf{S}} u_i^P (u_i^P)^\top$ with $u_i^P = \mathrm{diag}(\sqrt{P_k}) u_i^{\mathbf{S}}$ orthonormal (in Euclidean $\mathbb{R}^K$).

For $\mathcal{F}_{\mathrm{Eig}}$, we consider the case where

$$\lambda_1^{\mathbf{S}} > \lambda_2^{\mathbf{S}} > 0, \quad \lambda_i = 0 \text{ for } i \geqslant 3$$

$$(u_1^P)_i = \begin{cases} \sqrt{2/K} & \text{if } i \text{ is even} \\ 0 & \text{otherwise,} \end{cases} \quad (u_2^P)_i = \begin{cases} 0 & \text{if } i \text{ is even} \\ C \cdot i & \text{otherwise,} \end{cases}$$

where $C$ is a constant such that $\left\|u_2^P\right\| = 1$. Consider eigenvectors PEs with $q = 1$. The space $\mathcal{F}_{\text{Eig}}$ contains all the function $g$ of the form

$$g_i = f((u_1^{\mathbf{S}})_i) + f(-(u_1^{\mathbf{S}})_i)$$

for any $f$. By construction of $u_1^P$, these functions have the property of being 2-periodic: for all $i$, $g_i = g_{i+2}$. Now, $\mathcal{F}_{\mathbf{S}}(\mathcal{F}_{\text{Eig}})$ contains e.g. the following function

$$\mathbf{S}1 = KC_P 1 = \lambda_1^{\mathbf{S}}(1^\top u_1^P)u_1^P + \lambda_2^{\mathbf{S}}(1^\top u_2^P)u_2^P$$

This function is not 2-periodic: on $i$ odd, we have $(u_2^P)_i \neq (u_2^P)_{i+2}$ and therefore $g_i \neq g_{i+2}$, which concludes the proof.

For $\mathcal{F}_{\text{Dist}}$, taking again $q = 1$, we consider $K = 4$ and $C_{01} = C_{10} = 1$, and $C_{k\ell} = 0$ otherwise. The space $\mathcal{F}_{\text{Dist}}$ contains all the function $g$ of the form

$$g = \frac{1}{K}f(C)1$$

where $f$ is applied element-wise on $C$. With this choice, for any $f$ we have $g_3 = g_4 = f(0)$. However, the space $\mathcal{F}_{\mathbf{S}}(\mathcal{F}_{\text{Dist}})$ contains the function

$$h = \frac{1}{K^2}Cf(C)1$$

Here we have $h_3 = \frac{f(0)f(1)}{8} + \frac{7f(0)^2}{8}$ and $h_4 = f(0)^2$, which can be made unequal. Hence $h \notin \mathcal{F}_{\text{Dist}}$, which concludes the proof.

# C  Universality

Here we recall the universality results of [25], that were derived for an architecture called Structured GNN, in the case of non-random edges. In this paper, these results are valid for the distance-encoded PEs (10), through Theorem 3, our definition of $\mathcal{F}_{\text{Dist}}$ (11). The results in [25] basically proves that $\mathcal{F}_{\mathbf{S}}(\mathcal{F}_{\text{Dist}})$ satisfies the hypotheses of Prop. 3. We recall them here without proof. The following results are valid for adjacency matrix (ex. 1), distance-encoding PE (10), and SBM (ex. a) or p.s.d. kernel (ex. b), with other additional hypotheses in each cases.

- **SBM**: if $\mathcal{X}$ is finite, $C = [w(k,\ell)]$ is invertible, and $P$ is such that for all $s \in \{-1, 0, 1\}^K$, $s^\top P = 0$ implies $s = 0$, then $\mathcal{F}_{\mathbf{S}}(\mathcal{F}_{\text{PE}}) = L_{\sqcup}^2$.

- **Additive kernel**: if $w(x, y) = v(u(x) + u(y))$ with $u, v$ that are continuous and injective, then $\mathcal{F}_{\mathbf{S}}(\mathcal{F}_{\text{PE}}) = L_{\sqcup}^2$.

- **Unidimensional radial kernel**: if $\mathcal{X} = [-1, 1]$, $w(x, y) = v(|x - y|)$ with continuous injective $v$, and $P$ is symmetric (that is, $P([a, b]) = P([-b, -a])$ for all intervals), then $\mathcal{F}_{\mathbf{S}}(\mathcal{F}_{\text{PE}}) = L_{\sqcup}^2 \cap \mathcal{S}(\mathcal{X})$ where $\mathcal{S}$ are symmetric functions. If $P$ is not symmetric, then $\mathcal{F}_{\mathbf{S}}(\mathcal{F}_{\text{PE}}) = L_{\sqcup}^2$.

- **Spherical kernels**: If $\mathcal{X} = \mathbb{S}^{d-1}$ is the $d$-dimensional sphere, $w(x, y) = v(x^\top y)$ with continuous injective $v$, and $P$ has a density $p$ w.r.t. the uniform distribution on the sphere such that: the unique decomposition $p(x) = \sum_{k \geq 0} \sum_{j=1}^{N(d,k)} a_{k,j} Y_{k,j}(x)$ where $Y_{k,j}$ are spherical harmonics is such that $x \mapsto [\sum_{j=1}^{N(d,k)} a_{k,j} Y_{k,j}(x)]_k$ is injective (see [25] and references therein), then $\mathcal{F}_{\mathbf{S}}(\mathcal{F}_{\text{PE}}) = L_{\sqcup}^2$.

# D  Technical or third-party results

The following Lemma can be proved in a number of ways.

**Lemma 8** (Lemma 4 in [24]). *Let $\mathcal{X}$ be a compact metric space, and $\mathcal{Y}$ a measurable space. Consider a bivariate measurable function $U : \mathcal{X} \times \mathcal{Y} \to \mathbb{R}$ that is uniformly bounded, and continuous in the first variable. Let $y_1, \ldots, y_n$ be drawn i.i.d from a distribution $P$ on $\mathcal{Y}$. Then*

$$\left\| \frac{1}{n} \sum_i \eta(\cdot, y_i) - \int \eta(\cdot, y)dP(y) \right\|_\infty \xrightarrow[n \to \infty]{\mathcal{P}} 0$$

We extend the previous results to polynomials of the kernel.

**Lemma 9.** *For any bounded, continuous kernel $w_{\mathbf{S}}$, we have for all $k \geqslant 1$:*

$$\sup_{x,x'\in\mathcal{X}} \left| \frac{1}{n^k} \sum_{i_1,\dots,i_k} w_{\mathbf{S}}(x,x_{i_1})w_{\mathbf{S}}(x_{i_1},x_{i_2})\dots w_{\mathbf{S}}(x_{i_k},x') - (\mathbf{S}^{k+1}\delta_{x'})(x) \right| \xrightarrow[n\to\infty]{\mathcal{P}} 0 \qquad (35)$$

*Proof.* We prove it by induction. For $k = 1$, we have

$$\sup_{x,x'\in\mathcal{X}} \left| \frac{1}{n} \sum_i w_{\mathbf{S}}(x,x_i)w_{\mathbf{S}}(x_i,x') - \int w_{\mathbf{S}}(x,y)w_{\mathbf{S}}(y,x')dP(y) \right| \xrightarrow[n\to\infty]{\mathcal{P}} 0 \qquad (36)$$

by applying Lemma 8 on $\mathcal{X}\times\mathcal{X}$, and since $w_{\mathbf{S}}$ is continuous on a compact domain and therefore bounded and Lipschitz.

Then, assuming the property for $k-1$, we write

$$\sup_{x,x'\in\mathcal{X}} \left| \frac{1}{n^k} \sum_{i_1,\dots,i_k} w_{\mathbf{S}}(x,x_{i_1})w_{\mathbf{S}}(x_{i_1},x_{i_2})\dots w_{\mathbf{S}}(x_{i_k},x') - (\mathbf{S}^{k+1}\delta_{x'})(x) \right|$$

$$\leqslant \sup_{x,x'\in\mathcal{X}} \left| \frac{1}{n} \sum_{i_1} w_{\mathbf{S}}(x,x_{i_1}) \left[ \frac{1}{n^{k-1}} \sum_{i_2,\dots,i_k} w_{\mathbf{S}}(x_{i_1},x_{i_2})\dots w_{\mathbf{S}}(x_{i_k},x') - (\mathbf{S}^k\delta_{x'})(x_{i_1}) \right] \right|$$

$$+ \sup_{x,x'\in\mathcal{X}} \left| \frac{1}{n} \sum_{i_1} w_{\mathbf{S}}(x,x_{i_1})(\mathbf{S}^k\delta_{x'})(x_{i_1}) - \int w_{\mathbf{S}}(x,y)(\mathbf{S}^k\delta_{x'})(y)dP(y) \right|$$

The first part converge to $0$ using the boundedness of $w_{\mathbf{S}}$ and the recursive hypothesis, while the second converges to $0$ using again Lemma 8. $\qquad\square$

**Theorem 8** (Simplified Davis-Kahan, see [57])**.** *Let $A, \hat{A} \in \mathbb{R}^{d\times d}$ be symmetric with eigenvalues $\lambda_i$ and $\hat{\lambda}_i$ ordered by decreasing order and eigenvector $u_i$, $\hat{u}_i$. Take $p$ and assume that $\delta = \min(\lambda_{p-1} - \lambda_p, \lambda_p - \lambda_{p+1}) > 0$. Then there exists $s \in \{-1,1\}$ such that*

$$\|su_p - \hat{u}_p\| \leqslant \frac{\left\|A - \hat{A}\right\|}{\delta} \qquad (37)$$

**Theorem 9.** *Denote $W = [w(x_i,x_j)/n]_{ij}$, $\bar{W} = \left[ \frac{w(x_i,x_j)}{n\sqrt{d(x_i)d(x_j)}} \right]_{ij}$ and $L(M) = D_M^{-\frac{1}{2}}MD_M^{-\frac{1}{2}}$ with $D_M = \operatorname{diag}(M1)$*

*If $\alpha_n \gtrsim (\log n)/n$, then*

$$\left\| \frac{A}{\alpha_n n} - W \right\| \xrightarrow[n\to\infty]{\mathcal{P}} 0. \qquad (38)$$

*Moreover, if $d(x) \geqslant d_{\min} > 0$ and $\alpha_n \geqslant C(\log n)/n$ where $C$ is a constant that depends on $w$, then*

$$\|L(A) - \bar{W}\| \xrightarrow[n\to\infty]{\mathcal{P}} 0. \qquad (39)$$

*Proof.* The first result is due to Lei and Rinaldo [28].

For the second result, we have from [24, Theorem 6] that

$$\|L(A) - L(W)\| \to 0 \qquad (40)$$

Then, according to Lemma 8 we have that

$$\left\| d - \frac{1}{n} \sum_i w(\cdot,x_i) \right\|_\infty \xrightarrow[n\to\infty]{\mathcal{P}} 0 \qquad (41)$$

and in particular with probability going to one $d_i^W \stackrel{\text{def.}}{=} (D_W)_i \geqslant d_{\min}/2 > 0$ for all $i$.

Denoting by $\bar{D} = \operatorname{diag}(d(x_i))$, and noticing that $W = \bar{D}^{-\frac{1}{2}} W \bar{D}^{-\frac{1}{2}}$ and $\|W\| \leqslant 1$, we have

$$\left\| \bar{W} - L(W) \right\| = \left\| \bar{D}^{-\frac{1}{2}} W \bar{D}^{-\frac{1}{2}} - D_W^{-\frac{1}{2}} W D_W^{-\frac{1}{2}} \right\|$$

$$\lesssim \frac{1}{\sqrt{d_{\min}}} \left\| \bar{D}^{-\frac{1}{2}} - D_W^{-\frac{1}{2}} \right\| = \frac{1}{\sqrt{d_{\min}}} \max_i \left| \frac{1}{\sqrt{d(x_i)}} - \frac{1}{\sqrt{d_i^W}} \right|$$

$$= \frac{1}{\sqrt{d_{\min}}} \max_i \left| \frac{d(x_i) - d_i^W}{\sqrt{d(x_i) d_i^W} (\sqrt{d(x_i)} + \sqrt{d_i^W})} \right|$$

$$\lesssim \frac{1}{d_{\min}^2} \left\| d - \frac{1}{n} \sum_i w(\cdot, x_i) \right\|_\infty \xrightarrow[n \to \infty]{\mathcal{P}} 0$$

which concludes the proof. $\qquad\square$

