# OpenReview forum: "What functions can Graph Neural Networks compute on random graphs? The role of Positional Encoding"
_NeurIPS.cc/2023/Conference — NeurIPS 2023 poster_

### Official Review · Reviewer_Jsrn · 2023-06-23

**Soundness:** 4 excellent
**Presentation:** 3 good
**Contribution:** 3 good
**Rating:** 7
**Confidence:** 3

**Summary:**

This paper theoretically studies node classification on latent position random graphs, a generalization of graphons. More precisely, the family of functions that can be approximated using GNNs with positional encodings are determined in general and for the examples, sign-net (eigenvector PEs) and distance-based PEs.

**Strengths:**

* Very important and interesting topic. Most of current GNN theory focuses on WL and hence is mostly applicable to graph-level tasks. This paper takes a deep look on node classification.
* Uses (and apparently requires) significant novel techniques / bounds to achieve the results (e.g., USVT estimators, pp) of independent interest.
* Strong extension of [25]

**Weaknesses:**




Sometimes the paper is not fully self-contained, e.g.:
* the proper definition of graph shift matrices.
* more intuition on latent position random graphs and graphons.
* "special properties" of ReLU. Why don't you just say what properties you mean, Lipschitzness etc.


Minor:
* References are very badly maintained: A lot of references are missing either the Arxiv id or the venue (ICML, NeurIPS, ICLR etc.). Please fix this! The three references with arxiv ids are all published by now, their published versions should be used instead.
* Please cite also Morris et al., [2019] when you meantion the GNN vs WL connection (you only cite [51], while the papers essentially came out in parallel).
* sometimes you write "latent position random graphs" and sometimes "latent position*s* random graphs"
* the claim "as medium or large graphs are never isomorphic" is a bit of an exaggeration. Perhaps say "seldom" instead of never.
* While the authors explain in the text that $PE(S,Z)$ does not depend on $Z$, I wonder about the choice of notation. Wouldn't it be a lot clearer if one instead writes e.g., $PE(S)\oplus Z$ where $\oplus$ is concatation?
* The paper deals with node classification. However, it is written in a way such that the reader is left unsure whether the paper might deal with graph-level tasks or not. E.g., it's not mentioned in the abstract. Only at page 2 it is almost apologetically mentioned that the authors will focus on node classification "as this is makes more sense" on large graphs. Node classification is an important and interesting task on its own. Please mention that this is your subject of study clearly in the abstract and in the beginning of the paper.

Typo:
* Line 232 "that sum to 1" --> "that sums to 1"
* Line 59 "no not" --> "do not"


[Morris,  et al., 2019] "Weisfeiler and leman go neural: Higher-order graph neural networks." AAAI 2019

**Questions:**

1. I am a bit confused by Assumption 1 and that the authors claim that e.g., the adjacency matrix satisfies it. If I am not mistaken the operator norm of a matrix $A$ is its largest singular value. For adjacency matrices of graphs this is known to be lower bounded in terms of the (square root of) maximum degree in the graph. Hence $||S||$ seems to be unbounded even for simple (and sparse!) graphs such as trees. Only for graphs with upper bounded degree this assumption seems to hold. Please explain.
2. Can your results be generalized to general graph shift matrices, as defined by [46], i.e., matrices $S$ of size $n*n$ s.t. $S_{v,w}=0$ if ${v,w}$ is not an edge.
3. Theorem 2: What happens in the case of basis-invariance (and with non-distinct eigenvalues)?
4. Of course on some specific graphs GNNs will only be able to predict using the constant function (cliques, etc.). Still, in general the space of functions approximable by GNNs *without* PEs seems interesting and rich. Is there a reason that there is no theorem exactly determining the set of functions learnable by GNNs without PEs? I imagine that e.g., nodes in the same orbit (i.e., nodes that can be mapped by an automorphism onto each other) must habe the same function value, but besides that arbitrary functions are possible?

**Limitations:**

Please discuss the limitations of the latent position random graph model. On what kind of real-world data is it appropriate? Are there examples which cannot be modeled well with latent position random graphs?

---

> ### Author Rebuttal · Authors · 2023-08-07
>
> Thank you for your very detailed review.
>
> **Questions:**
>
> 1. In our analysis (Example 1) the adjacency matrix is normalized: $A/(\alpha_n n)$, which guarantees that its eigenvalues are bounded with high probability. It requires however the knowledge (or approximate knowledge) of the ``sparsity level'' $\alpha_n$, which is not the case for the normalized Laplacian, where the normalization is automatic.
>
> 2. For the first theorem, our results are valid as long as the convergence in Assumption 1 is satisfied, this can indeed include weighted edges depending on the choice. We will add the reference. For the results on PEs, we did the analysis for two representative examples, but they could also be extended on a case-by-case basis.
>
> 3. In this case, the convergence is known for so-called ``spectral projection'' operators, rather than individual eigenvectors (see eg Rosasco et al. [42]). For PEs in practice, the SignNet paper includes an architecture called BasisNet, which is used to handle eigenvalues with multiplicities by being basis-invariant (instead of sign-invariant). The expression is however quite more complex, and the convergence in the limit seems significantly more involved. We will add it to future work.
>
> 4. Indeed, the *limitations* of continuous-GNNs in the limit are not well understood, even without PEs. A simple example is that of constant-degree function, for which the output of continuous-GNNs with constant input remains constant (just as GNNs fail on degree-regular graphs). Some examples in [25] (which we generalize with PEs, see our Appendix C) include some simple symmetry analysis, but many questions remain open. We will focus on these limitations in future work.
>
> **Limitations:**
>
> We will add a more extensive introduction on latent positions random graphs in the available space. Such random graphs, generalizing SBMs and random geometric graphs, have been initially introduced for the analysis of social networks, but are also used for protein interaction networks, physical networks (road, computers), and many other examples. While very intuitive in their interpretation (latent variables can be made very concrete: user preferences, spatial positions, etc.), their drawbacks include mostly a limited capability to model degree power-laws or complex substructures. They remain however the most popular tool to model large graphs, with an active literature.
>
> **Other points/minor points**:
> - Thank you for pointing this out, we will fix the references, and add the suggested references
> - We will fix the terminology and notations as suggested

---

> > ### Comment · Reviewer_Jsrn · 2023-08-14
> >
> > Thank you for the clarifications.
> >
> > I still vote for accept and think that this paper would be valuable to the GNN community of NeurIPS.

---

### Official Review · Reviewer_2T44 · 2023-07-05

**Soundness:** 4 excellent
**Presentation:** 4 excellent
**Contribution:** 3 good
**Rating:** 7
**Confidence:** 3

**Summary:**

The paper provides a complete description of the function space generated by equivariant GNNs on latent position random graphs in order to study the expressive power of GNNs on large graphs for node level tasks. Then the paper investigates how node features such as positional encodings would affect the function space by studying two popular PE examples on two random graph models. The theoretical results suggest some normalization tricks that would help generalization across graphs of different sizes, and the paper validates this observation numerically in both synthetic and real data.

**Strengths:**

1. The paper studies the expressive power of GNNs by characterizing the function space generated by permutation-equivariant GNNs on latent position random graphs. As the paper argued in the paper, previous work on expressive power of GNNs has largely been focused on graph isomorphism testing, which is in fact a very limited framework when it comes to large graphs or node level tasks. The results established in the paper are much more relevant in those cases.

2. The question about what role does node features or positional encoding (PE) plays in graph learning is important, and has not been well-studied in the literature. This paper provides a novel way to study them outside the graph isomorphism framework, filling the gap between previous theoretical studies and the practice.
    - One particular weakness of results established under the graph isomorphism framework is that node features are often assume to be uniform/random/the unique node identifiers. This conditions are unrealistic and basically ignore the role of real node features play in learning on graphs.
    - While PE has been widely used empirically, to the best of my knowledge, theoretical understanding of it is very limited, particularly outside of graph isomorphism testing framework.

3. The paper is clearly written.

**Weaknesses:**

1. Although the objects of interest and technical results in this paper are mathematically precise, they are obscure.  It is hard to directly interpret the corresponding function space, and given a GNN architecture with a positional encoding, one cannot easily compute and compare the function space.

2. The results are established for random graph models. Although random graph models have been useful in helping gaining deeper insights into many graph learning problems, they are synthetic graphs and there are some real graph characteristics they cannot capture. For example, nodes in stochastic block models have similar degrees, while real graphs exhibit degree heterogeneity. It is hard to know how robust/relevant the results are to real graphs or other random graph models that are not studied in the paper.



**Questions:**

Please consider adding indices to the subplots and rewriting the caption for Figure 1, and place it around the same place as the relevant text. Currently sentence right after "From left to right" is very confusing to read.

---

> ### Author Rebuttal · Authors · 2023-08-07
>
> Thank you for your detailed and insightful review. We agree with the reviewer's opinion that many questions remain on the interpretability of the function spaces defined in this work, beyond the results presented in this paper. We will complete the outlook section if space permits. We also agree that random graph models are imperfect, however they remain the main tool available to describe large graphs in the limit. We note that, beyond SBMs, random graphs using for instance a Gaussian kernel are quite flexible (eg, in terms of degree function), especially since the distribution of latent variables $P$ is not constrained. To our knowledge, not much is known concerning other models such as the preferential attachment model, and convergence properties of operators such as the graph Laplacian are still open. The analysis is likely to be very different in this case, and this is an important path for future work.

---

> > ### Comment · Reviewer_2T44 · 2023-08-17
> >
> > Thank you for the response. I will keep my score. The theoretical development is nice and solid, but it needs better interpretability and more grounding in practice, as other reviewers mentioned.

---

### Official Review · Reviewer_rtyh · 2023-07-09

**Soundness:** 4 excellent
**Presentation:** 4 excellent
**Contribution:** 4 excellent
**Rating:** 6
**Confidence:** 1

**Summary:**

The paper investigates the theoretical expressiveness of Graph Neural Networks (GNNs) on large random graphs, and the role of node features and Positional Encodings (PEs) in enhancing their approximation power. The paper makes several contributions, such as:

- Providing a complete and intuitive description of the function space approximated by equivariant GNNs on random graphs, and showing that it is determined by a base set of input features and some extension rules;
- Analyzing two representative examples of PEs based on eigenvectors and distance-encoding, and extending previous universality results to more general random graph models;
- Proposing some normalization tricks to ensure convergence and consistency of PEs across different graph sizes, and demonstrating their effectiveness on synthetic and real data;
- Proving some new concentration inequalities for filtered random matrices that are of independent interest.

**Strengths:**

he paper is well-written and organized, and the technical proofs are detailed in the appendix or supplementary material. The paper also provides some interesting insights and discussions on the properties and limitations of GNNs and PEs. The paper deepens the theoretical understanding of GNNs and PEs, which are widely used and studied in various domains of machine learning on graphs.

**Weaknesses:**

N/A

**Questions:**

N/A

---

> ### Author Rebuttal · Authors · 2023-08-07
>
> Thank you for your review. We are delighted that you found our paper to be well-written and organized, with detailed technical proofs in the appendix.

---

### Official Review · Reviewer_mk3D · 2023-07-12

**Soundness:** 3 good
**Presentation:** 3 good
**Contribution:** 3 good
**Rating:** 7
**Confidence:** 4

**Summary:**

The paper analyzes the approximation capability of Graph Neural Networks (GNNs) in node-level tasks. The work presents a novel concept of expressivity that examines whether GNNs can approximate $L^2$-functions on measure spaces as the underlying graph grows and approximates operators on $L^2$ itself. The paper characterizes the precise class of functions that can be approximated and shows that positional encodings (eigenvector and distance-based) enhance the expressive power of GNNs in line with the new definition.


**Strengths:**

-    The authors give a new framework for investigating the expressivity of GNNs in node-level tasks.
-    They give mathematical results and proofs, enhancing the scientific rigor of the study.
-    The paper is well-written.
-    Beyond the GNN result, the concentration and universality results offer interesting and novel insights that enrich the paper.


**Weaknesses:**

-    The authors occasionally do not discuss related work or overstate their findings. For instance, they give a novel universality result on L^2 functions and assert that their normalization rules apply to real-world graphs. Nonetheless, they do not discuss related work and test the normalization only on two small datasets.
-    Similarly, they claim that their results are valid for general GNNs or MPNNs-like architecture, whereas the results are primarily applicable for GCN-like architectures with a normalized adjacency matrix or 1/n*A, which is not commonly used in practice.
-    The primary results and the majority of intermediate results in the appendix lack self-containment, which makes it challenging to follow the paper in these technical sections.
-    It's unclear why the broad applicability of graph matrices and operators S and $\mathbf{S}$ is necessary when the only examples given are the normalized adjacency matrix and the adjacency matrix normalized by the graph size itself.
-    The real-world experiments don't align with the theory as they involve graph-level predictions, but the expressivity measure introduced is at the node-level.

**Questions:**

 -   Can the authors elaborate why the 1-WL test or the concept of separating isomorphic nodes in the node-classification context is unsuitable for large graphs? Proposition 3 uses Stone-Weierstrass, indicating that distinguishing isomorphic graphs remains key for achieving universality.
 -   The SignNet architecture studied isn't commonly employed in practice. Can the authors comment on how the function (may) change if $f$ is another permutation equivariant function, such as GIN? This is interesting as GIN was used in the original paper.
  -  It would be beneficial to reference related works on universality results and concentration inequality, perhaps in the appendix.
- Regarding the enhanced expressive power, it would be good to discuss other works that demonstrate that positional encodings (PEs) indeed boost expressivity at the graph level or node level.
- Can the authors test the importance of normalizing PEs in node-level tasks? (see e.g. PPI datasets or [1])

## Minor Points:

   -  In Definition 1, the dependence on $n$ and what $S$ refers to is unclear. Generally one has to search too long for definition when reading the main results and results in the appendix.
   -  Out of interest: Are there any accessible condition under which $F_S(F_{\mathrm{PE}}) \subsetneq F_{S'}({F}_{{\mathrm{PE'}}})$ holds?
   -  There is a typo in line 594. It should be $\nu$ instead of $\varepsilon$.

## Conclusion:

This paper is a very nice and valuable read for anyone interested in GNNs and the utility of positional encodings (PEs) (from the perspective of large random graphs). Despite some overstated findings and certain ambiguity in the author's claims, the paper provides a new and valuable perspective on the expressivity of GNNs in the context of large random graphs. I recommend a score of 6, but am willing to increase my score if some issues are resolved.

[1] Dwivedi, Vijay Prakash, et al. "Long range graph benchmark." Advances in Neural Information Processing Systems 35 (2022): 22326-22340.

**Limitations:**

yes

---

> ### Author Rebuttal · Authors · 2023-08-07
>
> We thank the reviewer for their insightful and review.
>
> - **On related work and overstatement**: thank you for pointing this out, we will add more related work on universality (and other places where we feel it might be necessary). You are correct in pointing out that the normalization ``good practices'' are by-product of our analysis, that we do not expect to really improve large-scale state-of-the art models. We have performed more experiments in the global rebuttal, and will better frame the contribution.
>
> - **MPNN architecture**: apologies if our formulation is misleading, we indeed treat GNNs that use graph shift operators like the adjacency (Ex. 1) or normalized Laplacian (ex. 2), and did not claim to treat generic aggregation function. We will clarify this. You are also correct in noting that the normalization of the adjacency matrix $A/(\alpha_n n)$, necessary for convergence, is generally not used in practice. This is why the normalized Laplacian (or other degree-based normalization) is generally preferred.
>
> - **On operators notations**: we mainly treat two cases, the adjacency matrix (indeed globally normalized) in Example 1 and the symmetric normalized Laplacian in Example 2. These are quite different operators, that give rise to different operators $\mathbf{S}$ in the continuous case (see Ex. 1 and 2 in the paper). All of our results are valid for both, and some results more generally for any operators under some convergence assumption (Assumption 1). Hence we used generic notations.
>
> - **On experiments**: indeed, since our primary goal was to simply illustrate the normalization trick (on small models, with no claim to improve over state-of-the-art large models), we needed datasets with many graphs of very different sizes, which are far more common for graph-level tasks. We have however added three experiments for node-classification tasks, found in the global rebuttal.
>
> - **WL test**: What we meant is the following: while distinguishing (non-)isomorphic graphs are important for small or medium graphs like molecules (with dozens or hundreds of nodes, say), very large graphs have a very, very small chance of being exactly isomorphic, but they may present different characteristics for learning. For instance, two large graphs drawn from an SBM model have an exponentially small probability of being isomorphic, but they have the same number of communities, so it is interesting to characterize the fact that a GNN will indeed be able to approach a function that correctly labels the nodes in the limit. We precisely describe the space of such functions in this paper, as functions on the space of latent variables $\mathcal{X}$. We use the Stone-Weierstrass theorem in a different manner than graph-level discrete analyses: the space under consideration is not the space of discrete graphs (up to isomorphism), but the space of latent variables.
>
> - **SignNet and function change**: the SignNet architecture is a variant for *eigenvector-based PEs*, which are (to our knowledge) a quite common choice. SignNet resolves the sign ambiguity of eigenvectors, which is usually done by choosing signs randomly, an additional level of randomness that we felt would significantly complexify our analysis and notations. We will clarify the relation between the two and add more references to eigenvectors-based PEs. We are not entirely sure what function $f$ the reviewer is referring to? The input function to the GNN can be anything (degrees, etc.), as long as our convergence assumptions are satisfied. We chose two representative examples of PEs. We will clarify this point in the final version.
>
> - **More experiments**: Thank you for the suggestions. We found the PPI dataset, one of the only node-classif dataset to include several graphs of different size, to be quite peculiar: it seems almost only the nodes features are used, and PEs are usually ignored by trained models. LRGB datasets are difficult tasks, and most vanilla GNNs fail on them. We however added three experiments on node-classification, described in the global rebuttal.
>
> - **Reference and minor points**: thank you for the suggestions, we will add more references on universality and PEs. We will correct the typos and try to clarify our definitions. To our knowledge, the subset property must be checked on a case-by-case basis, but a more generic condition is an interesting path for future work, thank you.

---

> > ### Comment · Reviewer_mk3D · 2023-08-14
> >
> > Thank you for your thorough response. I have some additional questions:
> >
> > 1.    Could you include standard deviations in your experimental results?
> > 2.    Would you clarify the size of the synthetic node classification datasets you used?
> > 3.    Do the experiments actually employ the architectures analyzed, specifically SignNet and the distance-based PEs from [25]?
> > 4.    On lines 339 and following, you state: "On real data, we see that renormalization generally helps generalization...". Could you elaborate on this point? Since your work primarily analyzes expressivity, I'm curious whether you shatter the training datasets with and without PEs, and if only the generalization improves with (normalized) PEs. If this is the case, could you provide an explanation?
> > 5.    Regarding SignNet, the function $f$ in my review references equation (7) in your manuscript. Instead of an elementwise MLP, the original SignNet-paper incorporates GIN or other GNNs. If I am not mistaken this leads to certain expressivity results that do not hold for MLPs (e.g., approximation of other popular PEs, etc.)

---

> > > ### Author Response · Authors · 2023-08-16
> > >
> > > Thank you for the additional questions.
> > >
> > > 1. We will perform more runs and include the standard deviation. If we don't have time before the response period ends, in the final version.
> > > 2. Training set includes 800 graphs, whose sizes were drawn randomly from 70 to 600 nodes. In the "OOD" case, the test graphs have sizes ranging from 800 to 900.
> > > 3. We implement exactly the architecture analyzed. Compared to [25], our distance-PE additionally include an MLP filtering of $S$ (eq. (10)), in order to benefits from Theorem 4, (theoretically) necessary in our analysis.
> > > 4. Indeed, this is empirical observation, as we do not explicitely analyze generalization (nevertheless in our analysis, without normalization, we do not even have convergence on large graphs). Normalization helps when comparing PE with and without normalization. Without normalization, the PE values are not consistent from one graph size to the other (since, for instance, eigenvectors are normalized to $1$ in $\mathbb{R}^n$ for different $n$, they must be upscaled by $\sqrt{n}$ to be consistent from one graph to the other)
> > > 5. Indeed, we elected to keep the "message-passing" part separated from the computation of PEs for clarity, so considered only element-wise MLPs for SignNet, before feeding it to a GNN. Convergence would still apply with GIN within SignNet, as long as it uses a converging operator $S$ (eg normalized Laplacian). Indeed, some additional expressive power may result from it, we will add a clarifying sentence in the final version and maybe some results from the SignNet paper in Appendix for completeness.

---

> > > > ### Comment · Reviewer_mk3D · 2023-08-17
> > > >
> > > > Thank you for the response. Assuming the described changes are indeed implemented, I believe this is a valuable contribution and a nice read. Well done! I've raised my score to 7.

---

### Official Review · Reviewer_gCL6 · 2023-07-20

**Soundness:** 3 good
**Presentation:** 4 excellent
**Contribution:** 2 fair
**Rating:** 5
**Confidence:** 4

**Summary:**

This paper aims to enhance the theoretical understanding of Graph Neural Networks (GNNs) concerning their expressive power, focusing on large graphs.  The paper provides a comprehensive description of the function space generated by equivariant GNNs for node tasks, incorporating notions of convergence from previous examples. The role of input node features is emphasized, along with the study of node Positional Encodings and equivariant eigen-based features. The theoretical findings suggest that certain normalization tricks positively impact GNN generalization on synthetic and real data, supported by new concentration inequalities introduced in the paper.

**Strengths:**

* The paper is very well written and was very enjoyable to read.

* The analysis is fairly novel and studies GNNs with powerful features.

* The positioning of the paper with respect to existing literature is clear and honest.

**Weaknesses:**

* The connection between the derived analysis and the practical implementation of GNNs is not clear.

* The paper suggests the message-passing operations might not add up to the representation power of the GNN output features, which is not very well supported. For example, the eigenvectors and eigenvalues of the graph shift operator are able to perfectly reconstruct the graph structure. The analysis suggests that equivariant eigen-based features, which are less expressive compared to eigenvectors, not to mention eigenvectors and eigenvalues combined, do not benefit from message-passing operations. Current analysis suggests the opposite.

* The analysis is mainly based on random graphs, which admit certain regularities, that do not allow concrete conclusions about the representation power of GNNs.

* There are certain claims that are not properly supported, e.g., "In some context, however, WL-based analyses have limitations: they pertain to tasks on graphs (e.g., graph classification or regression) and have limited to no connections to tasks on nodes or links; and they are mostly relevant for small-scale graphs, as medium or large graphs are never isomorphic but exhibit different characteristics."

How is the size of the graph related to graph isomorphism? What do the authors mean by small-scale graphs, medium or large graphs?

**Questions:**

* Please address the above concerns.

* I would appreciate it if the authors make a direct connection between their analysis and practical implementations of GNNs. How is this work benefits deep learning on real graphs?

* I would also expect some more detailed experiments with respect to the benefit of the proposed normalization.

**Limitations:**

The limitations of the work are discussed in the previous sections. I am willing to raise my score if my comments are addressed properly.

---

> ### Author Rebuttal · Authors · 2023-08-07
>
> We thank the reviewer for their detailed and insightful review.
>
> - **Practical implementation**: the main goal of this paper is to better understand the theoretical properties of existing GNNs on large graphs, and we focus on classical implementations that use the adjacency matrix or normalized Laplacian, as well as classical PEs. We offer some (limited) recommendations of practical implementation: the small remark on normalization good practices, as well as the use of normalized Laplacian instead of adjacency matrix, since the latter requires additional knowledge of the sparsity level $\alpha_n$. We also prove universality results for some choices of PEs. However we agree that leveraging this type of analysis to develop truly different, more ``powerful'' GNNs (in the infinite-node limit) remains a major avenue for future work.
>
> - **Message-Passing and PEs**: the eigenstructures are able to reconstruct the graph only when considering the full $n$ eigenvectors/values, which is not the case here (we cut at some finite $q$). The full spectrum is actually impossible in our analysis, since we let $n$ go to infinity. Our analysis does not really suggest that message-passing is not ``useful'', as it extends the space of available functions from $F_{PE} $ to $F_S(F_{PE})$: indeed, Prop. 6 even proves the contrary in some cases. The characterizations of these spaces is case-dependent however, and we agree that there are still many open questions on a case-by-case basis. We will add a clarifying sentence.
>
> - **Random graphs**: random graphs do indeed admit certain regularities. However they are, as of now, one of the only tools available to statistically model "large" graphs, which is necessary when we want to model large-scale phenomenon such as communities or manifold structure. The resulting analysis is indeed very different from the usual discrete combinatorial one (which may be what the reviewer meant by "concrete conclusions about the representation power of GNNs"?), but we see them as complementary analyses rather than competing ones.
>
> - **Isomorphism**: thank you for pointing this out, we will clarify the sentence in the final version. What we meant is the following: while distinguishing (non-)isomorphic graphs are important for small or medium graphs like molecules, large graphs have only a very, very small chance of being exactly isomorphic. However, they may present different characteristics for learning (eg communities), and it is these characteristics that we aim to modellize with random graphs. For instance, two large graphs drawn from an SBM model have an near-zero probability of being isomorphic, but they have the same number of communities, so it is interesting to characterize the fact that a GNN will indeed be able to approach a function that correctly labels the nodes in the limit. We precisely describe the space of such functions in this paper.
>
> - **More experiments**: as discussed in the paper, the proposed normalization is a by-product of our analysis that we do not expect to bring significant performance boost to state-of-the-art large architectures (that are able to learn such normalization within the range of graph sizes in the training set), but is an interesting fact nonetheless. We have however performed more experiments on synthetic data and the CiteSeer node classification dataset (modified to have several graphs of different sizes), described in the global rebuttal.
>
> We hope to have addressed your concerns as stated in the Limitations section, and that you will be able to raise the proposed score.

---

> > ### Comment · Reviewer_gCL6 · 2023-08-17
> > **I am raising my score**
> >
> > I would like to thank the authors for their response. I am raising my score to 5. I cannot raise it to higher scores, since there are still concerns about the practicability of this analysis.

---

### Official Review · Reviewer_2UtV · 2023-07-25

**Soundness:** 4 excellent
**Presentation:** 2 fair
**Contribution:** 2 fair
**Rating:** 6
**Confidence:** 3

**Summary:**

In this paper the authors provide characterisations of the class of functions that can be approximated by GNNs, possibly in the presence of positional encodings (PE). Here the notion of convergence is stated with high probability and for increasing graph size. So, in large graph limit.

**Strengths:**

- There is a lot to be liked in this paper since it provides a rigorous analysis of the function spaces that can be approximated.
- The technical statements seem correct to me.
- Presentation in the beginning of the paper is fine, but section 4 is rather dense.
- The problem considered is quite natural, if one is interested this kind of limiting behaviour of graph learning methods.
- Recent, state-of-the-art PE approaches are considered.
- Giving a bit of hints in the main paper as how things are proved is appreciated.

**Weaknesses:**

- Although familiar with theoretical expressiveness results of GNNs, this material was new for me. It took some time to understand what was meant by "GNNs become functions in the limit". Once this was clear, I looked for insights as to what results about the limit situation could tell about capabilities of concrete practical GNN architectures. More effort could have been made to make the aim and reason why more accessible.
- On line 57, it is mentioned that [25] does not consider random edges, but it does. This is a bit confusion. It is also mentioned the current paper generalises previous results from [25], but this could have been made more precise.
- Throughout the paper I was looking for connections to approximation properties of GNNs (with PE) on finite graphs. It is well known that approximation capabilities are closely related to the distinguishing power of GNNs (and PE), and under conditions not unlike those in Definition 2, universality wrt functions with matching distinguishing power can be shown (see e.g., work by Azizian and Lelarge). Most of the result resemble this to some extent. A discussion on more classical approximation results in this context.
- As mentioned earlier, section 4 is rather dense and the presentation could be improved. For example, what are are the precise assumptions on eigenvalues and vectors that you assume in Section 4.2.1? Also you define function classes in the theorems  without much explanation. It makes reading and understanding the result more difficult than it should.
- The results in section 4 seem to depend on SBM or PSD kernels. It is unclear why this assumption is made.


**minor**
l6. certains functions -> certain

**Questions:**

- Please elaborate on what kind of practical insights one can get from the analyses. In the discrete finite case, approximation results are easier to interpret and use to build "better" models.
- A gut feeling tells me that there must be a close connection to approximation properties in the finite case. Since, PEs for example, add to the distinguishing power and hence approximation capabilities. Could you comment?
- Could you intuitively explain why SBM and PSD kernel assumptions are necessary in Section 4?


**Limitations:**

Not applicable.

---

> ### Author Rebuttal · Authors · 2023-08-07
>
> We thank the reviewer for their detailed and insightful review.
>
> - **Practical insight**: as mentioned by the reviewer, on large (random) graphs, in the infinite-node limit, node-task GNNs approach functions. Approximation results indicate *which labelling functions the GNN is able to approximate*, e.g., will it be able to distinguish communities in an SBM in the limit with high probability -- something that is not directly reflected by *discrete* approximation power results. It is however true that, as of now, the *limitations* of GNNs have been less clearly identified than in the discrete case, where combinatorial reasonings *à la* WL yields obvious limitations. Hence the scarcity, for now, of ``better'' models built in the limit, but also the interest of a better understanding of these objects.
>
> - **Connection with finite case**: Indeed, thank you for pointing this out. Some (as of now, limited) connections can be made with results in the discrete case: for instance, in the absence of PEs, with constant input, GNNs will have a constant output on degree-regular graphs, just as in the limit they will have a constant output function on random graphs with constant degree function. Beyond this, it seems than GNNs on large graphs are less ``limited'' than in the discrete case (see first point), but on the other hand universality results are only known with PEs. A better connection with discrete analyses is definitely a major avenue for future work. As suggested, we will add a discussion in the final version.
>
> - **SBM/psd assumptions**: These assumptions are not necessary *per say*, but sufficient. Indeed, the strong convergence of PEs such as eigenvectors does not seem to be true for a generic kernel $W$ (to the best of our knowledge, this is still an open question), and some mild regularity assumptions on $W$ are generally made. We focused on two representative, very popular models of random graphs, for which the proofs are reasonable for all our examples of PEs and graph operators. On a case-by-case basis, other models may of course be studied in the future. We will add a clarifying sentence in the final version.
>
> - **Other remarks**: We will improve the connection with [25], as well as the readability of the theorems in Sec. 4.

---

> > ### Comment · Reviewer_2UtV · 2023-08-11
> > **Reply to rebuttal**
> >
> > I have read the rebuttal and thank the authors for their feedback. Based on my own review and reading the other reviews, I am going to stick to my current score. The theoretical development is nice and solid, but it really needs more grounding in practice, possibly by relating it to finite case settings, or more widely used mpns's.

---

### Author Rebuttal · Authors · 2023-08-07

Dear reviewers,

We thank you for your valuable comments. We are glad to see that see that our paper is well-received and considered as *well-written* (gCL6, mk3D, rtyh, 2T44). The novelty is highlighted by several of you (gCL6, mk3D, 2T44, Jsrn).

A common concern of the referees is the practical insights provided by our manuscript and the connection with existing discrete analyses. We detail below additional experiments, and address the latter point, and other comments, in each individual rebuttal.

We provide three node-classification experiments (in addition to IMDB-BINARY and COLLAB). Several referees emphasized that our analysis focused on node classification tasks, but that we implemented our numerical illustrations of the normalization trick on graph classification datasets. As mentioned in the paper, to illustrate the effect of normalization, the dataset needs to include several graphs, and the vast majority of datasets for node classification have only one large graph. One of the only classical node-classif dataset with several graphs, namely PPI, is challenging for our purpose: most vanilla GCNs learn to ignore PEs entirely and use almost exclusively node features. In the absence of node features, they generally fail to learn. Hence, we used a synthetic dataset (using sklearn's ``make_classification`` to create the $x_i$ then a Gaussian kernel $W$ to create the random graph, a strategy also adopted by the Pytorch Geometric library), and another "real-world" dataset created by *extracting many subgraphs* from a node-classification dataset that had originally only one large graph, such as Citeseer. Additionally, for synthetic data we test the effect of having test graphs significantly larger than training graphs, which we denote by OOD (out-of-distribution), which emphasize the importance of proper normalization for generalization.

Note that moreover, we forgot to mention in the paper that for all experiments (including IMDB-BINARY and COLLAB), we do not use the original node features, but only the PEs computed from the graph structure, to emphasize the effect of the PEs. This detail will be added.

These rebuttal experiments may be further refined for the final version.

|                   | Eigen w/norm | Eigen w/o norm | Dist w/norm | Dist w/o norm |
|-------------------|--------------|----------------|-------------|---------------|
| Synthetic         | 68.61        | 65.59          | 67.31       | 62.49         |
| Synthetic OOD     | 67.87        | 62.51          | 66.80           | 63.33             |
| Citeseer-subgraph | 49.45        | 49.43          | 48.99       | 37.09         |

---

### Decision · Program_Chairs · 2023-09-21

**Decision:**

Accept (poster)

**Comment:**

The paper studies and generalizes known results regarding the expressive power of equivariant GNNs on random graphs, including special attention to the role of positional encoding in that setup. All reviewers liked the paper and supported acceptance.

Some criticism was raised about the relation of this paper and its results to practical GNN architectures and also the relation to previous expressive power results on smaller graphs. I believe that the paper would greatly benefit from a deeper discussion of these relations.